# Characterization of p38α autophosphorylation inhibitors that target the non-canonical activation pathway

Lorena González[1], Lucía Díaz [2], Joan Pous [1], Blazej Baginski [1], Anna Duran-Corbera[1], Margherita Scarpa[1], Isabelle Brun-Heath [1], Ana Igea[1], Pau Martin-Malpartida [1], Lidia Ruiz[1], Chiara Pallara[2], Mauricio Esguerra [2], Francesco Colizzi[1,6], Cristina Mayor-Ruiz[1], Ricardo M. Biondi[3], Robert Soliva[2], Maria J. Macias [1,4,7] ✉, Modesto Orozco [1,5,7] ✉ & Angel R. Nebreda [1,4,7] ✉

p38α is a versatile protein kinase that can control numerous processes and plays important roles in the cellular responses to stress. Dysregulation of p38α signaling has been linked to several diseases including inflammation, immune disorders and cancer, suggesting that targeting p38α could be therapeutically beneficial. Over the last two decades, numerous p38α inhibitors have been developed, which showed promising effects in pre-clinical studies but results from clinical trials have been disappointing, fueling the interest in the generation of alternative mechanisms of p38α modulation. Here, we report the in silico identification of compounds that we refer to as non-canonical p38α inhibitors (NC-p38i). By combining biochemical and structural analyses, we show that NC-p38i efficiently inhibit p38α autophosphorylation but weakly affect the activity of the canonical pathway. Our results demonstrate how the structural plasticity of p38α can be leveraged to develop therapeutic opportunities targeting a subset of the functions regulated by this pathway.

p38α, also known as MAPK14, is a ubiquitously expressed mitogen-activated protein kinase (MAPK) family member that has a key role in maintaining cellular homeostasis and tissue organization. The p38α structure is topologically similar to other MAPKs and consists of two lobes linked by a flexible hinge, with the junction between the two lobes forming the ATP binding site. The N-terminal lobe is composed of β-sheets, whereas the C-terminal lobe is mostly helical[1]. The activation of p38α normally involves phosphorylation by the upstream kinases MKK3 and MKK6, but p38α can be also activated by non-canonical mechanisms that involve autophosphorylation[2]. One of these mechanisms implicates the scaffold protein TAB1 (transforming

growth factor-β-activated protein kinase 1 (TAK1)-binding protein 1), which binds to p38α increasing its affinity for ATP and triggering its autophosphorylation in cis[3,4]. The binding of TAB1 to p38α induces a conformational change of the αC-helix, which facilitates ATP binding and the rearrangement of the activation loop. This allosteric mechanism of non-canonical activation has been associated with cardiomyocyte death in diseases such as myocardial ischemia-reperfusion injury, which ultimately leads to heart failure[5–10].

The evidence supporting a causal role for the kinase activity of p38α in several pathologies has led to the generation of a substantial number of chemical compounds, which inhibit p38α. Most of these

[1]Institute for Research in Biomedicine (IRB Barcelona), The Barcelona Institute of Science and Technology, 08028 Barcelona, Spain. [2]Nostrum Biodiscovery, 08034 Barcelona, Spain. [3]Instituto de Investigación en Biomedicina de Buenos Aires (IBioBA)-CONICET-Partner Institute of the Max Planck Society, Buenos Aires, Argentina. [4]ICREA, Pg. Lluís Companys 23, 08010 Barcelona, Spain. [5]Departament de Bioquímica i Biomedicina, Facultat de Biologia, Universitat de Barcelona, 08028 Barcelona, Spain. [6]Present address: Department of Marine Biology and Oceanography, Institute of Marine Sciences ICM-CSIC, 08003 Barcelona, Spain. [7]These authors jointly supervised this work: Maria J. Macias, Modesto Orozco, Angel R. Nebreda. ✉e-mail: maria.macias@irbbarcelona.org; modesto.orozco@irbbarcelona.org; angel.nebreda@irbbarcelona.org

compounds, such as the pyridinyl imidazole SB203580[11] and the N-aryl pyridinone PH797804[12], directly compete with the binding of ATP. Other inhibitors like the diaryl urea compound BIRB796[13] can bind to an allosteric pocket near the ATP-binding site and interfere with p38α activation by stabilizing the DGF-out inactive conformation, thereby impairing its phosphorylation by upstream kinases. Unfortunately, all these inhibitors have shown limited efficacy in clinical trials for long-term treatments, and none of them has progressed to phase III, fueling the interest in discovering new compounds with alternative inhibitory mechanisms[14]. In this line, several publications have reported compounds that bind to p38α in areas other than the ATP-binding site, stabilizing inactive conformations of the kinase or affecting the phosphorylation of specific substrates such as MK2[15–18]. Alternatively, targeting non-canonical p38α activation has been proposed as a potential therapeutic approach for cardiovascular pathologies[19]. The strategies to target this pathway mostly focus on interfering with the interaction between p38α and TAB1 by using either cell permeable peptides[20] or small-molecule compounds derived from adamantanes, which reduce TAB1 phosphorylation but do not impair p38α autoactivation[21]. Other alternatives have explored the application of hetero-bifunctional small molecules that target p38α for degradation[22,23].

Here, we report compounds that block p38α autophosphorylation in vitro but only slightly reduce the ability of p38α to phosphorylate substrates. We have also characterized by X-ray crystallography how these molecules interact with p38α. This study shows that these molecules interact with p38α in different regions around the active cavity, including the active site itself. We provide evidence that these p38α inhibitors reduce cardiomyocyte death in cellular models of ischemia-reperfusion, suggesting their potential applications as lead molecules to explore new treatments for cardiac dysfunctions.

## Results

### Identification of small molecules that inhibit non-canonical p38α activation

Inspired by the abundant structural information on small molecules that bind to different sites in the p38α protein, as well as by the existence of ligands with more than one binding site[21,24,25], we conducted an in silico high-throughput screening campaign to identify p38α ligands. We screened approximately 2.5 million low-molecular-weight compounds from the ZINC12 database[26] and retrieved 35 compounds that showed the most promising binding profile (see "Methods"). We decided to focus on potential regulators of non-canonical p38α activation, and analyzed the ability of the selected compounds to modulate TAB1-induced p38α autophosphorylation in vitro (Fig. 1a). The results allowed us to identify several compounds that inhibited p38α autophosphorylation to different degrees, with compound NC-27 showing the strongest inhibition of TAB1-induced p38α activation (Fig. 1b and Supplementary Fig. 1). Next, we selected additional commercially available compounds with similar chemical structures, and also designed and synthesized a number of analogs in an effort to improve the drug-like properties of the molecules (see "Methods"). Altogether we tested 98 additional compounds in the p38α autophosphorylation assay. In total, out of the 133 compounds analyzed, we identified 23 that displayed greater than 80% inhibition of TAB1-induced p38α autophosphorylation (Fig. 1c).

It has been reported that non-canonical p38α activation induced by TAB1 binding occurs during myocardial ischemia-reperfusion injury and contributes to triggering cardiomyocyte death in vitro and in vivo[9,10,20,27]. Thus, we investigated whether our most active compounds could interfere with cardiomyocyte death in a well-established in vitro model of simulated ischemia-reperfusion (SIR)[8,28]. H9c2 cells were treated for 2 h with simulated ischemia followed by 4 h of reperfusion, and apoptotic cell death was measured by the detection of cleaved caspase-3 (Fig. 2a). In agreement with previous reports[29–31],

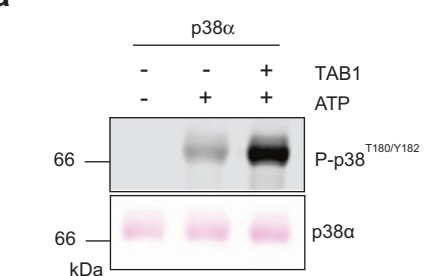

a

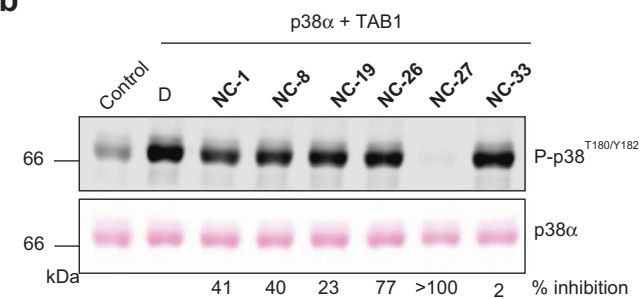

b

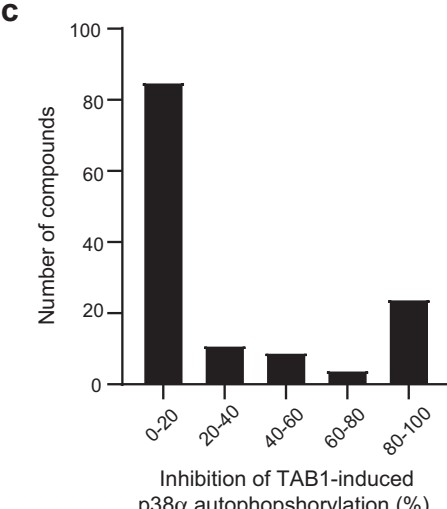

c

**Fig. 1 | Identification of compounds that inhibit TAB1-induced p38α autophosphorylation.** **a** Purified GST-p38α protein (1.5 μM, 2 μg in 20 μl) was incubated in autophosphorylation buffer with TAB1$_{386–414}$ peptide (15 μM) and ATP (600 μM), as indicated. After 2 h at 37 °C, samples were analyzed by Ponceau staining and immunoblotting. Results are representative from $n = 3$ experiments. **b** Purified GST-p38α was incubated with TAB1$_{386–414}$ peptide and ATP in the presence of DMSO (D) or the indicated compounds at 30 μM, and were processed as in (**a**). Control, GST-p38α, and ATP without TAB1. Results are representative from $n = 3$ experiments. **c** Histogram showing the average inhibitory activity of the 133 compounds analyzed at ≥10 μM in the TAB1-induced p38α autophosphorylation assay. Source data are provided as a Source data file.

treatment with the ATP-competitive inhibitor of p38α SB203580 abolished the appearance of cleaved caspase-3 in SIR-treated H9c2 cells. Notably, compounds NC-37 and its fluorinated analog NC-38 (Supplementary Fig. 1), substantially decreased also the cleaved caspase-3 levels (Fig. 2a). Then, we analyzed SIR-treated H9c2 cells by flow cytometry using Annexin V/ Propidium Iodide (PI) staining, and confirmed that compounds NC-37 and NC-38 were able to reduce SIR-triggered cell death to a similar extent as SB203580 (Fig. 2b). In addition, we observed a reduction in SIR-induced cell death upon TAB1 downregulation (Fig. 2c), which is consistent with the role described

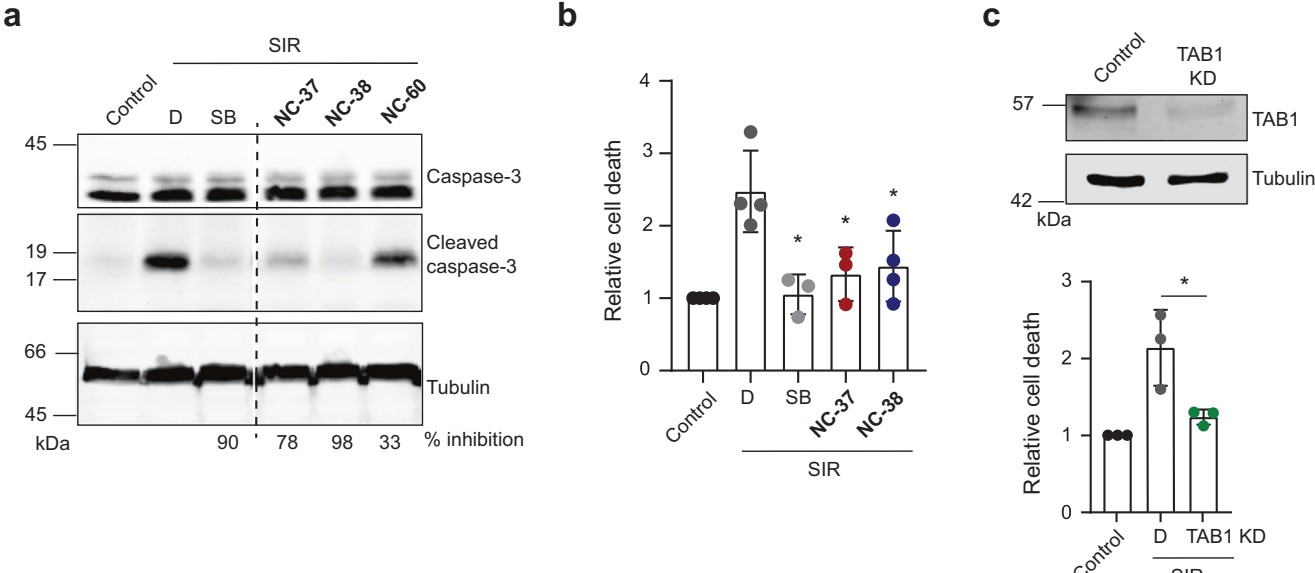

**Fig. 2 | Identification of compounds that inhibit ischemia-reperfusion-induced cell death. a** H9c2 cells were untreated (Control) or treated for 2 h with simulated ischemia followed by 4 h of reperfusion (SIR), in the presence of DMSO (D), the ATP-competitive inhibitor SB203580 (10 μM) or the indicated compounds at 30 μM. The compounds were added 24 h before and maintained during the treatment. Total cell lysates were analyzed by immunoblotting. Cleaved caspase-3 levels were quantified and normalized to DMSO-treated cells to calculate the percentage of inhibition. Results are representative from $n \geq 3$ experiments for compounds **NC-37**, **NC-38**, and **NC-60**. All the samples were analyzed in the same immunoblotting membrane and the dashed line indicates removed samples. **b** H9c2 cells were treated with the indicated compounds as in (**a**) and then were subjected to SIR, stained with Annexin V/Propidium Iodide (PI), and analyzed by flow cytometry. Cell death values were standardized giving the value of 1 to the control in each experiment. Data are shown as the mean ± SD (Control, D and **NC-38**, $n = 4$; SB and **NC-37**, $n = 3$). Two-sided unpaired t-test was used to compare the relative cell death in samples treated with the different inhibitors versus D. *$p < 0.05$. **c** H9c2 cells were untreated or transfected with TAB1 siRNA (TAB1 KD) and then were subjected to SIR as in (**a**). Cells were stained with Annexin V/PI and analyzed by flow cytometry to determine cell death values, which were standardized giving the value of 1 to the control in each experiment. Data are shown as the mean ± SD ($n = 3$ experiments). Two-sided unpaired t-test was used to compare the relative cell death in TAB1 KD versus D. *$p < 0.05$. TAB1 downregulation was analyzed by immunoblotting using total cell lysates. Source data are provided as a Source data file.

for TAB1 as a non-canonical activator of p38α during cell death induced by SIR[5]. Therefore, we selected for further characterization compounds **NC-37** and **NC-38**, which are both derivatives from the initial hit **NC-27** (Supplementary Fig. 1).

## Compounds NC-37 and NC-38 are highly selective inhibitors of p38α autophosphorylation

We investigated the ability of compounds **NC-37** and **NC-38** (Fig. 3a) to inhibit TAB1-induced p38α autophosphorylation in vitro at different concentrations, and found that both compounds have a half-maximal inhibitory concentration (IC50) within the low micromolar range (Fig. 3b). We next tested whether these compounds also impaired the TAB1-induced p38α activation in cells[3]. HEK293T cells were co-transfected with vectors encoding GFP-TAB1 and myc-p38α, treated with compounds **NC-37** and **NC-38**, and then p38α phosphorylation was analyzed by immunoblotting. In agreement with the in vitro assays, both compounds significantly decreased TAB1-induced p38α phosphorylation levels in a dose-dependent manner (Fig. 3c).

Since p38α can spontaneously autophosphorylate in the absence of TAB1[5], we investigated the effect of compounds **NC-37** and **NC-38** on this particular activity. We observed that both compounds strongly reduced the intrinsic ability of p38α to autophosphorylate (Fig. 4a and Supplementary Fig. 2a). However, neither compound **NC-37** nor **NC-38** affected the canonical activation of p38α by the MAP2K MKK6[32] (Fig. 4b). Next, we evaluated the effect of these compounds on the phosphorylation of substrates by MKK6-activated p38α. We found that the kinase activity of p38α on the substrates MK2 and ATF2 was only partly reduced by compounds **NC-37** and **NC-38** at 10 μM, while a strong inhibitory effect was observed with the ATP-competitive inhibitor PH797804 at 2 μM (Fig. 4c and Supplementary Fig. 2b).

We also tested the ability of compounds **NC-37** and **NC-38** to inhibit the activation of the p38α pathway in vivo, by using cell lines treated with stress stimuli, which are thought to engage mostly the canonical activation pathway (Supplementary Fig. 2c). We noticed that, as previously reported[33], the ATP competitor SB203580 partially reduced p38α activation, which might reflect the interference of SB203580 with the phosphorylation of p38α by MAP2Ks[34]. Importantly, using phosphorylation of the p38α substrate MK2 as a readout, we found that SB203580 completely abrogated the canonical activation of the p38α pathway in the three cell lines tested, but compounds **NC-37** and **NC-38** consistently showed a milder inhibitory effect, with H9c2 cells apparently being more sensitive to these compounds than U2OS and BBL358 cells (Supplementary Fig. 2d).

To investigate the specificity of these compounds, we first tested their ability to inhibit the catalytic activity of 97 human protein kinases, and found that compounds **NC-37** and **NC-38** at 10 μM were both highly selective towards p38α, and to a lesser extent also inhibited p38β and COT (Fig. 5a). By performing in vitro kinase assays using purified recombinant proteins, we found that compounds **NC-37** and **NC-38** had a substantially reduced inhibitory potency on the basal autophosphorylation of p38β compared to p38α (Supplementary Fig. 2e). Next, we measured the ability of compound **NC-38** to displace a chemical probe that binds to the active site of protein kinases, and found that p38α was the only target reproducibly detected out of 468 human protein kinases tested (Fig. 5b, c).

Taken together, our results show that compounds **NC-37** and **NC-38** are highly selective inhibitors for p38α non-canonical activation, having a milder effect on its ability to phosphorylate substrates. Therefore, we named these compounds NC-p38i for non-canonical p38α inhibitors.

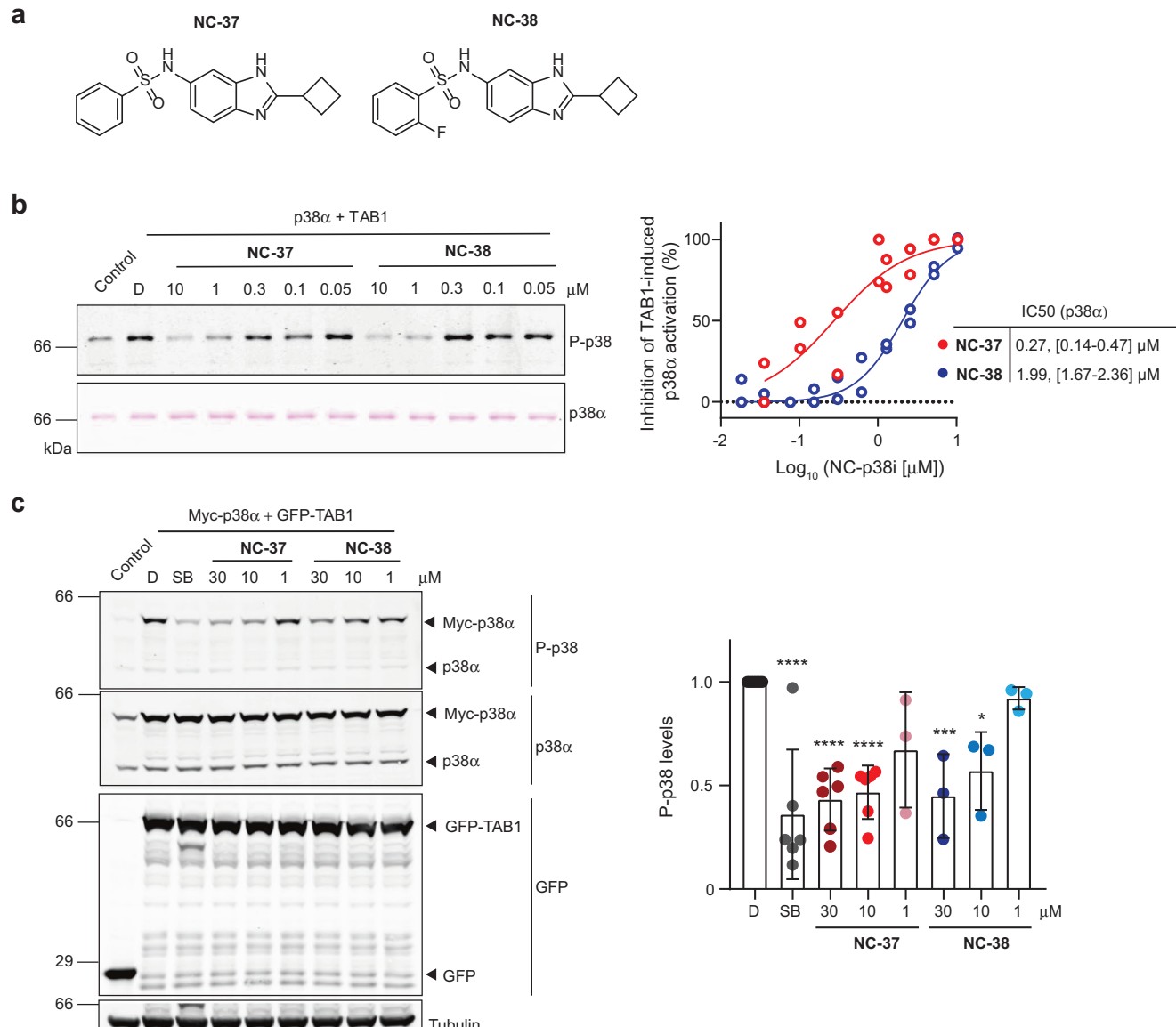

**Fig. 3 | NC-p38i compounds inhibit TAB1-induced p38α autophosphorylation.**
**a** Structure of compounds **NC-37** and **NC-38**. **b** Purified GST-p38α protein (1.5 µM, 2 µg in 20 µl) was incubated in autophosphorylation buffer with TAB1$_{386-414}$ peptide (15 µM) and ATP (600 µM) in the presence of DMSO (D) or the compounds **NC-37** and **NC-38** at the indicated concentrations. Control, GST-p38α and ATP without TAB1. After 2 h at 37 °C, samples were analyzed by Ponceau staining and immunoblotting with antibodies anti-phospho-p38 (T180/Y182). Values were normalized to the phosphorylated p38α levels in the DMSO sample. Data were fitted using a nonlinear regression fit model (Graphpad Prism) to determine the IC50s, with 95% confidence intervals indicated in brackets. $n = 2$ experiments. **c** HEK293T cells were

co-transfected with GFP-TAB1 and myc-p38α and then were treated with SB203580 (SB, 10 µM), DMSO (D) or compounds **NC-37** and **NC-38** at the indicated concentrations for 6 h. Control, cells co-transfected with GFP and myc-p38α. The histogram shows the quantification of p38α phosphorylation levels normalized to the DMSO-treated cells. Data are shown as the mean ± SD (D, $n = 8$; SB 30 µM and **NC-37** 10 µM, $n = 6$; **NC-37** 1 µM and all concentrations of **NC-38**, $n = 3$). One-way ANOVA test was used to compare the P-p38 levels in samples treated with the different inhibitors versus D. *$p < 0.05$, ***$p < 0.001$, and ****$p < 0.0001$. Source data are provided as a Source data file.

## NC-p38i do not interfere with TAB1 binding to p38α

Since TAB1 binding to p38α has been shown to induce conformational rearrangements that facilitate its autophosphorylation, we first used fluorescence polarization (FP) assays to investigate if the NC-p38i compounds affect the binding between TAB1 and p38α. Non-phosphorylated p38α was incubated with a TAB1 peptide (amino acids 386–414), which has been reported to bind to p38α[6]. The TAB1$_{386-414}$ peptide was N-terminally labeled with fluorescein isothiocyanate (FITC) and binding to p38α was monitored by measuring the FP of FITC. In agreement with published data[6], we observed a dose-dependent increase in fluorescence after the addition of p38α, which indicated its binding to FITC-TAB1$_{386-414}$ ($K_D = 0.9889 \pm 0.16$ µM)

(Supplementary Fig. 3a). For competition assays, a p38α concentration that produced an 80% increase in FP was incubated with increasing concentrations of NC-p38i. We observed that neither **NC-37** nor **NC-38** decreased the FP signal, suggesting that they did not compete with TAB1 interaction to p38α (Fig. 6a).

We also performed thermal shift assays (TSA) to measure the melting temperature (Tm) of p38α incubated with NC-p38i compounds either alone or together with the TAB1$_{386-414}$ peptide. Denaturation of non-phosphorylated GST-p38α protein revealed the sequential unfolding of both p38α and the GST tag (Supplementary Fig. 3b). We used SB203580 as a positive control, which strongly stabilized p38α as shown by a large shift of the GST-p38α unfolding curve

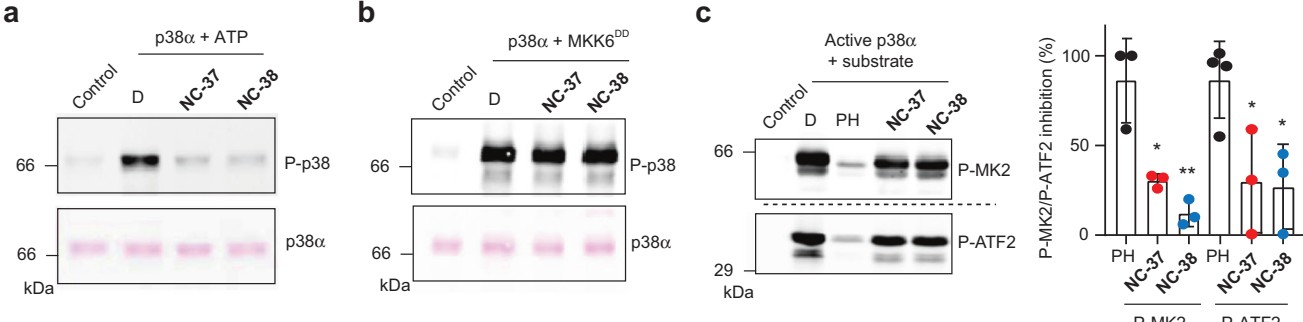

**Fig. 4 | NC-p38i compounds inhibit p38α autophosphorylation but not the canonical pathway. a, b** Purified GST-p38α (2 µg) was incubated with ATP (600 µM) (**a**) or with constitutively active MKK6[DD] (0.5 µg) (**b**) in 20 µl of kinase buffer and in the presence of DMSO (D) or the compounds **NC-37** and **NC-38** (10 µM). After 30 min at 37 °C, samples were analyzed by Ponceau staining and immunoblotting with antibodies anti-phospho-p38 (T180/Y182). Control, GST-p38α without ATP (**a**) or MKK6 (**b**). Results are representative from $n = 3$ experiments. **c** GST-p38α activated by MKK6 as in (**b**) (200 ng) was incubated with the substrates GST-MK2 or GST-ATF2 (0.5 µg) in 20 µl of kinase buffer and in the presence of DMSO (D),

PH797804 (PH, 2 µM) or the compounds **NC-37** and **NC-38** (10 µM). After 30 min at 30 °C, samples were analyzed by immunoblotting with antibodies against MK2 phospho-T334 or ATF2 phospho-T71. Control, GST-MK2 or GST-ATF2 without active GST-p38α. The histogram shows the percentage of phosphorylation inhibition using only the stronger band for the quantification in both cases. Results are shown as mean ± SD from $n = 3$ experiments. Two-sided unpaired t-test was used to compare the P-MK2/P-ATF2 inhibition in samples treated with the different inhibitors. $*p < 0.05$ and $**p < 0.01$. Source data are provided as a Source data file.

(Supplementary Fig. 3b, c) in agreement with published data[18,35]. The stabilizing effect of NC-p38i was moderate and dose-dependent, as reflected by a modest shift to the right of the GST-p38α denaturation curves (Fig. 6b), with compound **NC-37** stabilizing GST-p38α slightly better than **NC-38** (Supplementary Fig. 3c). Control experiments performed in parallel using the GST protein alone showed that neither SB203580 nor NC-p38i compounds shifted the denaturation curves of the GST tag (Supplementary Fig. 3d), confirming that NC-p38i compounds bind to p38α in vitro.

Interestingly, the combination of NC-p38i compounds with the TAB1[386-414] peptide resulted in higher Tm shifts of p38α compared with the NC-p38i alone (Fig. 6b). These observations strongly suggest that NC-p38i and TAB1[386-414] can bind simultaneously and to distinct sites of p38α.

### NC-p38i bind to the endogenous p38α in cells

We performed TSA in intact cells to test whether NC-p38i compounds could bind to the endogenous p38α protein, which is normally forming complexes with other proteins such as MK2[36]. Cells were treated with compounds **NC-37** or **NC-38** or with the vehicle DMSO, and then heated in a temperature range between 39 °C and 52 °C to monitor protein unfolding and aggregation. We observed that incubation with NC-p38i stabilized p38α, as shown by the higher amount of p38α protein remaining in the soluble fraction of cells heated at 45 and 48 °C compared with DMSO-treated cells (Fig. 6c). These results further support that NC-p38i compounds can bind to the endogenous p38α in cells.

### Structural characterization of NC-p38i binding to p38α

To characterize at an atomic level the interaction between NC-p38i and non-phosphorylated p38α, we performed co-crystallization experiments with either compound **NC-37** or **NC-38** and with or without the ATP competitive inhibitor SB203580 or the slow hydrolyzable ATP analog ATP-γ-S. Both SB203580 and ATP-γ-S have been reported to bind to p38α facilitating its crystallization[37]. As for p38α, we used a recombinant protein that contains the single point mutation C162S, which facilitates crystallization by reducing its aggregation[38].

We obtained well-diffracting crystals for three complexes of p38α with **NC-37** or **NC-38**, two in the absence of co-factors and one with SB203580 bound to the ATP binding site. The structures confirmed the main features of the p38α fold (Fig. 7a), as verified by the low Root Mean Square Deviation (RMSD) values of their superposition to that of

the previously characterized C162S mutant (PDB 1R3C): 7Z9T:0.891, 7PVU:0.837, 7Z6I:0.931. These structures revealed that NC-p38i compounds occupy three different areas around the active cavity, including the active site itself as well as the hinge pocket (behind the active site) and the allosteric pocket[13] (Fig. 7b). We did not observe any additional electron density in our maps that could be attributed to ligands. In the crystal with SB203580, which binds with nanomolar affinity to the active site, compound **NC-37** that exhibits micromolar affinity was prevented from accessing the active center and localized in the hinge region interacting with H48, K79, L87, T106, H107, K165, and Q355 side chains (PDB 7Z6I, Fig. 8a). In the absence of SB203580, the NC-p38i molecules occupy the active site, forming an extensive network of contacts independently of the presence or absence of the fluorine atom. The amino acids involved in the interactions were very similar in both compounds, with residues V30, A51, K53, L104, T106, L108, M109, L167, and D168 forming the cavity that accommodates the ligands (Fig. 8b). Interestingly, in the chain B of the 7Z9T structure, the ligand bound to the allosteric pocket adjacent to the active site, where the sulfonamide moiety of compound **NC-37** was at interacting distance from residues K53, L171, Y35, and D168. In addition, one nitrogen in the benzodiazol ring of the compound interacted with R67 (Fig. 8c).

All the solved structures were unphosphorylated, with the DFG motif at the beginning of the activation loop in the active conformation (DFG-in). We observed that the Asp side chain of the DFG motif was oriented towards the ATP binding site and participated in direct interactions with the compounds (Fig. 8d). In line with most of the available p38α crystal structures, the activation loop was flexible and it was not traceable in our complexes. It has been reported that TAB1-induced p38α autophosphorylation requires the formation of an intramolecular hydrogen bond between T185 and D150. This interaction enables the orientation of the activation loop towards the catalytic site, and constitutes an essential event for the p38α autophosphorylation process[7]. Interestingly, this hydrogen bond was absent in our three structures, and the helix comprising residues 182–188 (where T185 is located) was also poorly formed (Fig. 9). Therefore, the absence of this short helix and the hydrogen bond probably contributes to the selective inhibition of p38α autophosphorylation observed upon NC-p38i binding.

The amino acid Y35 on the P-loop has been reported to be highly dynamic and display two different orientations known as In and Out, respectively, depending on whether its aromatic ring aligns parallel to the main chains of S32 and G33 (Y35-in) or towards the protein's

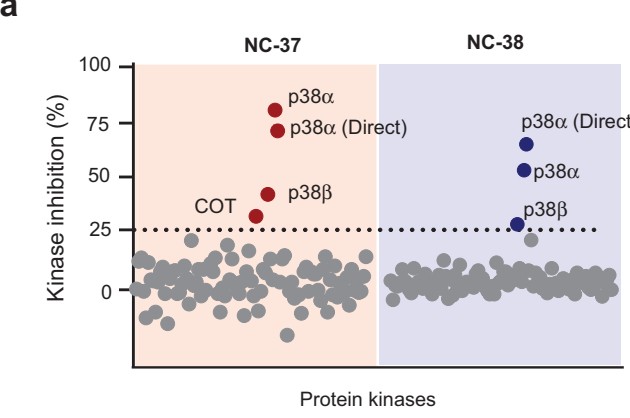

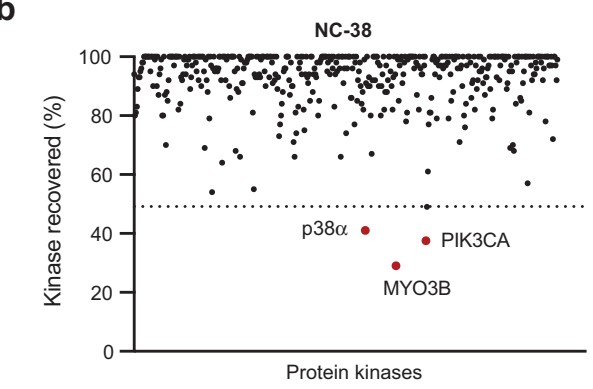

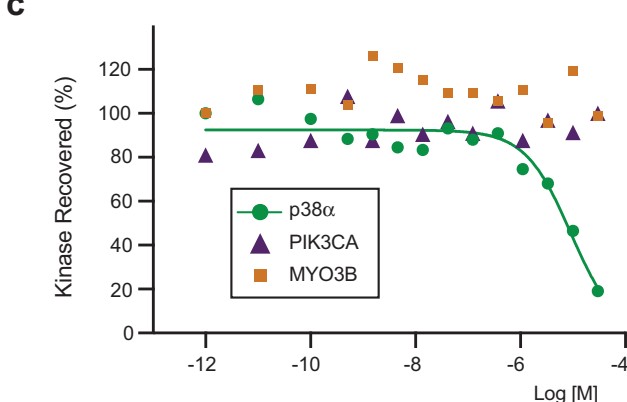

**Fig. 5 | NC-p38i compounds are specific for p38α. a** Percentage of inhibition on the catalytic activity of 97 human protein kinases incubated with compounds **NC-37** or **NC-38** (10 μM). Kinases were assayed at the ATP concentration of their apparent Km or below. Two different protocols were used to assay p38α kinase activity referred to as p38α and p38α (Direct). Results are shown as the mean of $n = 1$ done in duplicates. **b**, **c** The ability of compound **NC-38** (10 μM) to displace a chemical probe that binds to the active site of 468 human protein kinases was analyzed (**b**). The three kinases that show less than 50% recovery in the presence of **NC-38** were re-analyzed (**c**) in duplicates and using a range of concentrations, and only the binding to p38α was confirmed. Source data are provided as a Source data file.

α$_C$-helix (Y35-out)[39]. In the published complexes of p38α with SB203580 (PDB 1A9U, 3GCP), the Y35 aromatic ring is positioned towards the methylsulfinylphenyl group of the inhibitor[40,41], similar to the Y35-in orientation. In agreement with these observations, in the 7Z6I structure, SB203580 is bound to the active site through interactions with Y35, V38, A51, K53, L104, T106, L108, M109, and D168 side chains, and we also observed the Y35-in conformation (Supplementary Fig. 4). Intriguingly, we only observed Y35-out orientations in the complexes of p38α with

NC-p38i molecules (Supplementary Fig. 4), suggesting that this conformation is the most favorable to enable access of NC-p38i to the active site of p38α.

## NC-p38i behave as weak ATP-competitive compounds
The observation that NC-p38i can bind to the active site of p38α suggests that these compounds could compete with ATP binding. Our crystal structures show that NC-p38i do not displace the inhibitor SB203580 bound to non-phosphorylated p38α, but were able to bind to the catalytic site region in crystals obtained with the non-hydrolysable ATP-γ-S analog and in the absence of SB203580. To explore the idea that NC-p38i might bind to the non-phosphorylated p38α more efficiently than ATP, we investigated the effect of increasing ATP concentrations on the ability of NC-p38i to inhibit p38α autophosphorylation in vitro. We observed that compounds **NC-37** and **NC-38** were both able to inhibit p38α autophosphorylation with similar IC50s at 100 and 600 μM ATP, whereas higher ATP concentrations (2 mM) decreased their inhibitory potency (Fig. 10a). These results support that NC-p38i may compete with the binding of ATP to non-phosphorylated p38α.

Next, we tested if NC-p38i compete with ATP binding to p38α in cells by performing NanoBRET kinase target engagement[42]. Cells expressing NanoLuc-fused p38α were incubated with a cell-permeable energy transfer probe, whose binding to the active site of p38α generates a BRET signal due to its proximity to the NanoLuc fluorescent tag. As a positive control, we used the ATP-competitive inhibitor VX-702, which efficiently decreased the BRET signal (Fig. 10b). Interestingly, we found that compound **NC-37** was able to displace the active site-binding probe and lowered the BRET signal but at higher concentrations than VX-702 (Fig. 10b). These experiments further support that NC-p38i can inhibit ATP binding to p38α, but less efficiently than the classical ATP-competitive inhibitors such as VX-702. The results are consistent with the ability of these compounds to bind to several regions in and around the ATP-binding pocket of p38α, nevertheless showing a remarkable specificity for this protein kinase.

To account for the specificity of NC-p38i, we compared the ATP-binding cavities of p38α and related protein kinases. Sequence alignments confirmed that key residues involved in ATP-binding are strictly conserved in p38 family members, but the surrounding residues are more variable, affecting the size and charge distribution of the ATP-binding pocket (Supplementary Fig. 5a, b). Similar differences were observed at both the hinge pocket and the allosteric binding site (Supplementary Fig. 5c, d). Thus, while p38α and p38β have similar pockets, regarding both shape and electrostatic properties, p38γ and p38δ are different, corroborating our observation that NC-p38i only inhibit p38α and p38β. When compared to other MAPKs, we observed that only ERK2 could potentially interact with NC-p38i based on the cavity size but the charge distribution was different (Supplementary Fig. 5c, d).

## Discussion
Most p38α inhibitors that have been developed over the past two decades were designed to target the ATP binding site[11,12]. Some of these compounds are very potent and rather specific, but their ability to inhibit p38α in many contexts, including homeostatic functions, probably contributes to the toxicity detected, which ends up compromising their therapeutic efficacy. Alternative strategies being explored to target p38α include inhibitors tailored to particular diseases that are applied locally[14], inhibitors that are used in combination with chemotherapy drugs for short-term treatments[43,44], or inhibitors that target specific substrates such as MK2[15,17]. Emerging evidence suggests that targeting the non-canonical pathway of p38α activation may be relevant in some cases, given that this activation mechanism is restricted to a subset of functions in specific cell types[5,45]. Since the non-canonical

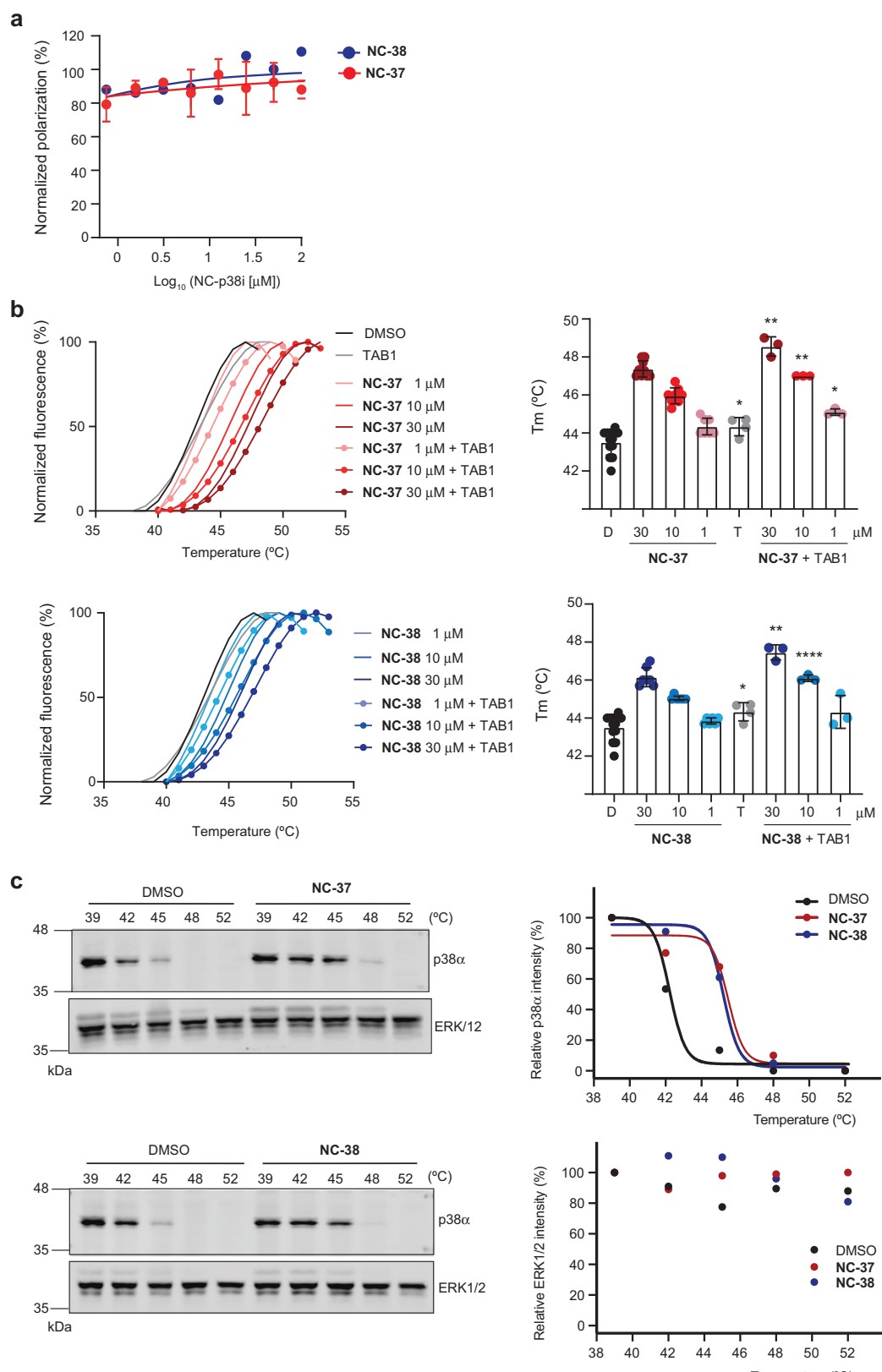

activation pathway relies on p38α autophosphorylation, understanding the mechanisms that underlie this process is crucial to design targeted therapeutic strategies. Here, we report the identification of p38α inhibitors referred to as NC-p38i, which efficiently inhibit the autophosphorylation of p38α while having a minor effect on the canonical p38α pathway.

Experiments showing that NC-p38i can displace a chemical probe that binds to the ATP site of p38α both in vitro and in cells suggest that these compounds can engage the active site of p38α, although NC-p38i possess lower affinity for p38α than the typical ATP-competitive inhibitors. When they bind to the catalytic site, they select the non-phosphorylated DFG-in p38α conformation. Furthermore, we have

**Fig. 6 | NC-p38i compounds bind to and stabilize p38α without inhibiting its interaction with TAB1. a** FITC-labeled peptide TAB1[386-414] (10 nM) was incubated for 1 h with purified GST-p38α (5.6 µM) and with increasing concentrations of compounds **NC-37** and **NC-38** (0.75–100 µM), as indicated. The FP signal was monitored as a readout of protein interaction. Experiments were done in triplicates, and results are shown as mean ± SD (**NC-37**, $n = 3$) or the mean (**NC-38**, $n = 2$). **b** Representative denaturation curves of GST-p38α in the presence of compounds **NC-37** and **NC-38** at the indicated concentrations alone or in combination with the TAB1[386-414] peptide (15 µM). DMSO was used as control. Curves were generated by normalizing fluorescence values, considering the lowest value as 0 and highest value as 100% in each condition. Results are representative from $n = 3$ experiments, except for **NC-38** 1 and 30 µM ($n = 2$). The histograms (right panels) show the calculated Tm values. Two-sided unpaired t-test was used to compare the Tm of GST-p38α incubated with TAB1 (T) versus DMSO control (D), and **NC-37** or **NC-38** alone versus **NC-37** or **NC-38** combined with TAB1. Results are shown as mean ± SD (D, $n = 13$; **NC-37** 30 µM, $n = 9$; **NC-37** 10 µM, $n = 8$; **NC-37** 1 µM, and **NC-38** 30 µM and 10 µM, $n = 7$; **NC-38** 1 µM, $n = 6$; T, $n = 4$; all concentrations of **NC-37** and **NC-38** + TAB1, $n = 3$). *$p < 0.05$, **$p < 0.01$ and ****$p < 0.0001$. **c** KBM7 cells were treated with DMSO or compounds **NC-37** and **NC-38** (30 µM) for 2 h and then heated in a temperature range from 39 °C to 52 °C to induce protein unfolding and aggregation. Cell lysates were analyzed by immunoblotting, and the indicated bands were quantified and represented in the graphics. $n = 1$. Source data are provided as a Source data file.

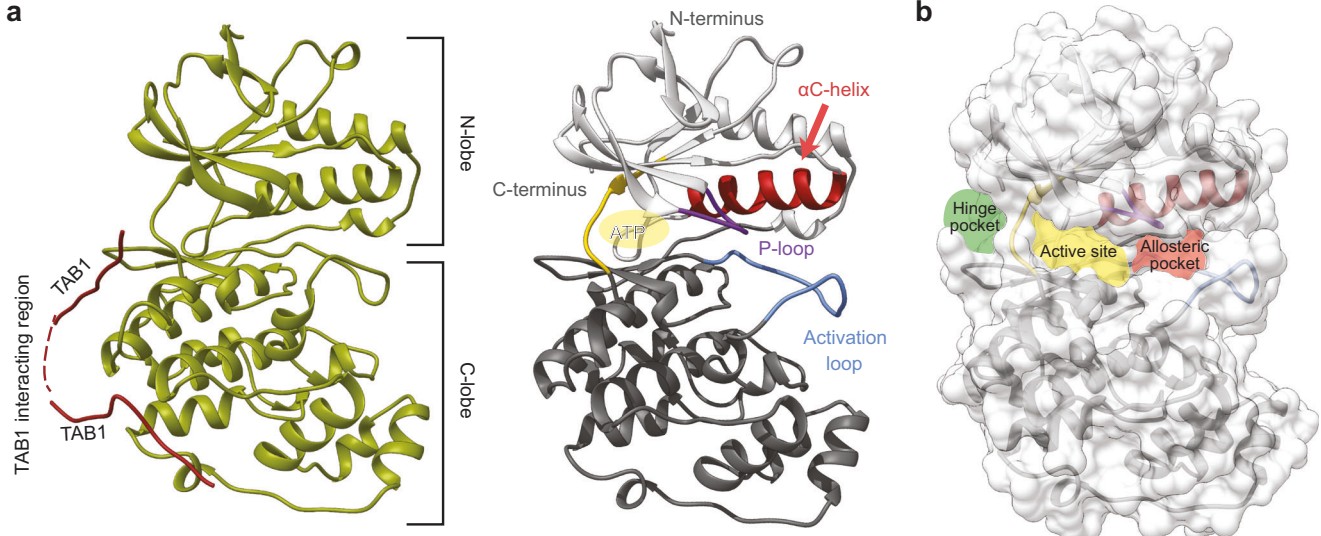

**Fig. 7 | Schematic representation of p38α structures. a** Complex of p38α and TAB1 (PDB:4LOO) and a representative p38α structure determined in this work (PDB:7Z6I) displaying the different structural and functional regions characteristic of p38α, and indicating the ligand binding sites occupied by TAB1 and the ATP binding region highlighted in yellow. **b** The three binding pockets of NC-p38i identified in the crystals are represented on the p38α surface.

found that the side chain of Y35 does not participate in the recognition of NC-p38i molecules, and is oriented outside the ATP-binding region, in contrast to the orientation observed in complexes with SB203580 (Y35-in). In addition, we found that NC-p38i can also bind to two other regions near the catalytic site of p38α, the so-called hinge pocket and the allosteric pocket. This alternative binding mode was observed in crystals that either have SB203580 bound to the active site of p38α or were obtained in the presence of ATP-γ-S. Interestingly, we could not observe in the crystal structures of p38α bound to NC-p38i the hydrogen bond between T185 and D150, which was previously identified as crucial for p38α autophosphorylation[7]. Therefore, it seems that the three binding sites of NC-p38i compounds interfere with the formation of this key interaction, which leads to the inhibition of p38α autophosphorylation but barely affect the canonical p38α signaling. Of note, although NC-p38i bind to the ATP pocket and nearby regions of p38α with micromolar affinity, they do not seem to target other protein kinases, at least in vitro. This striking specificity could be accounted for by the size and charge distribution of the ATP-binding region in p38α, which is different from other MAPKs. Taken together, the specificity of NC-p38i for p38α combined with their ability to inhibit autophosphorylation but not the transphosphorylation activity, supports the potential pharmacological interest of these compounds.

The observation that NC-p38i can inhibit the intrinsic p38α autophosphorylation, which is TAB1-independent, suggests that the primary mechanism of action of these compounds is not to interfere with TAB1 binding. In agreement with this idea, our results indicate that NC-p38i do not inhibit the binding of TAB1 to p38α, and support

that TAB1 and NC-p38i can possibly bind simultaneously to the non-phosphorylated p38α structure. It seems likely that p38α autophosphorylation involves some conformational rearrangements, which spontaneously happen during basal p38α autophosphorylation and are probably boosted by TAB1 binding. Hence, NC-p38i may bind to the active site and/or nearby regions of a prone-to-autophosphorylate p38α conformation, probably interfering both with ATP binding and with the conformational changes that are required for p38α autophosphorylation, including the hydrogen bond formation between T185 and D150[7]. This would explain why NC-p38i preferentially inhibit both TAB1-induced and spontaneous p38α autophosphorylation but do not alter the canonical pathway, as determined by the MAP2K-catalyzed phosphorylation of p38α and its ability to phosphorylate substrates. In the non-phosphorylated p38α, whose affinity for ATP is probably very low[39,46], NC-p38i compounds can compete with ATP for binding to the active site, as observed in crystals obtained in the presence of the non-hydrolysable ATP-γ-S, and NC-p38i binding involves residues such as K53, M109, and D168, which are critical to form key interactions with ATP. However, once p38α is phosphorylated in its activation loop and in the presence of a substrate, its affinity for ATP increases substantially[39,46], which is likely to impair the binding of NC-p38i to the active site and hence reduce their ability to inhibit the catalytic activity of p38α. Taken together, our results provide an explanation for how compounds with moderate affinity like NC-p38i can inhibit non-canonical p38α activation without affecting the canonical pathway, and support that these compounds can modulate p38α autophosphorylation-associated cellular processes.

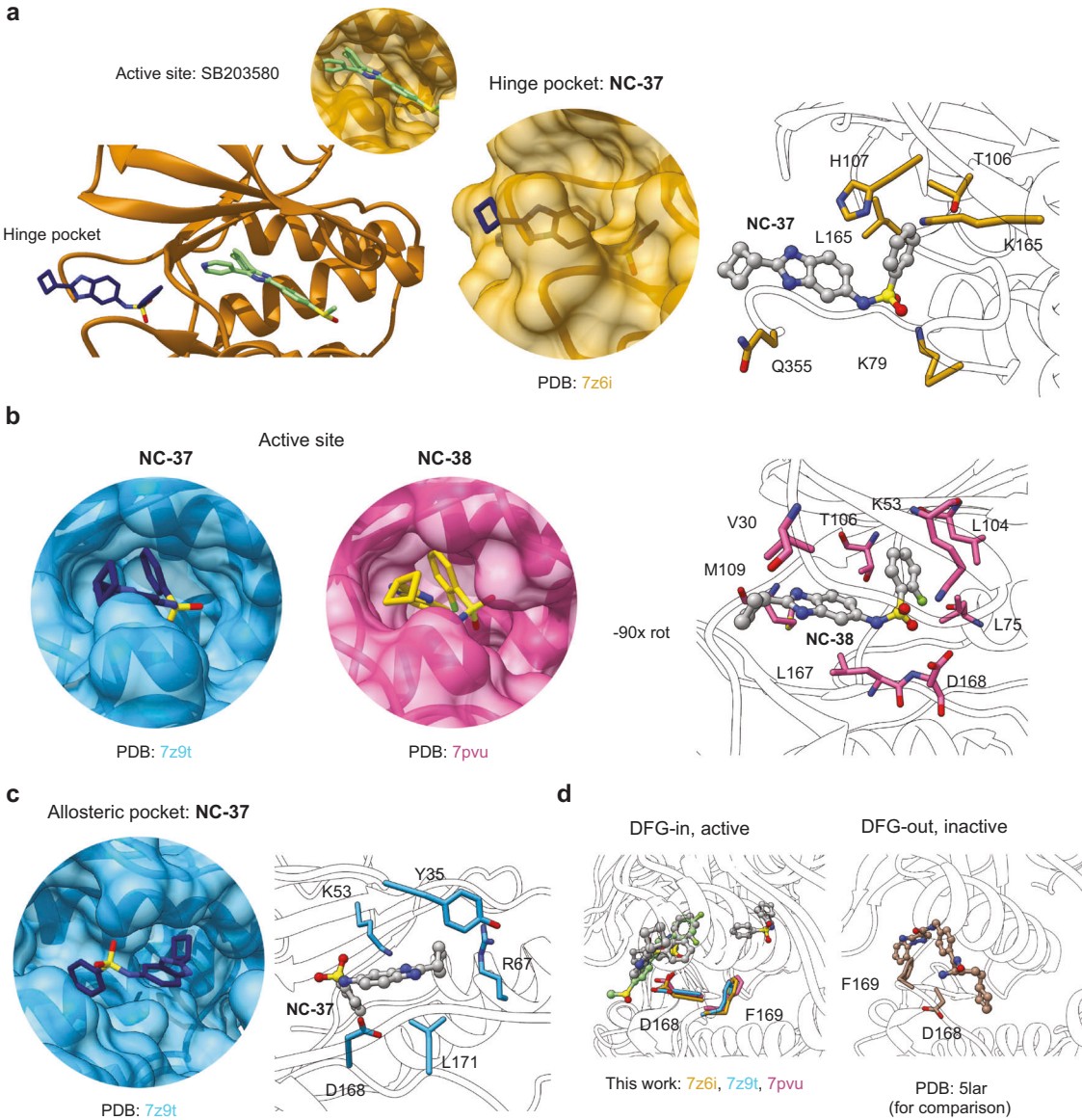

**Fig. 8 | Structural characterization of the binding of NC-p38i compounds to p38α. a** Crystal structures of p38α in complex with SB203580 (located in the active site) and compound **NC-37** bound to the hinge pocket. **b** p38α complexes bound to **NC-37** (dark blue, PBD:7Z9T) and **NC-38** (yellow, PBD:7PVU) bound to the active site. Both complexes are highly similar. Specific residues involved in the interaction are shown for the **NC-38** complex (compound represented in gray, the fluorine atom is shown in green). **c** The p38α complex bound to **NC-37** (chain B) shows the compound bound to the allosteric pocket adjacent to the active site. Residues involved in the binding are highlighted. The X-ray data collection and refinement statistics are shown in Supplementary Table 4. **d** All complexes of p38α with NC-p38i show the DFG motif in the active conformation (DFG-in). Compounds are labeled in the figure for clarity.

TAB1-induced non-canonical p38α activation has been implicated in pathological situations of cardiomyocytes and endothelial cells, suggesting that the inhibition of this non-canonical pathway could be useful to treat some diseases[45]. Supporting this hypothesis, our results show that NC-p38i can decrease cell death in a cellular model of SIR. Further in vivo experiments will be needed to evaluate whether these compounds may provide the basis to develop future therapeutic agents for cardiovascular diseases. Importantly, the ability of NC-p38i to inhibit TAB1-independent p38α autophosphorylation in vitro, suggests that these compounds might also modulate other processes that have been reported to involve non-canonical p38α activation. For example, the T-cell receptor activation in CD4+ T cells induces the phosphorylation of p38α on Y323 followed by its autophosphorylation[47], which is important for infiltrating T cells to produce cytokines and promote the progression of pancreatic tumors[48]. This suggests that inhibitors of p38α autophosphorylation could be used to ameliorate tumor growth.

In summary, the ability of NC-p38i compounds to effectively inhibit p38α autophosphorylation by weakly binding to several regions in and around the ATP-binding pocket of p38α, combined with their specificity for this protein kinase, reveals unique properties of these compounds that open a new approach for the identification of multi-binding site drugs with pharmacological interest.

## Methods

### In silico screening

In order to find small molecules that potentially inhibit p38α autophosphorylation, a wide exploration of promiscuous binding sites was carried out using the unphosphorylated inactive p38α (PDB:4LOO). Different grid boxes were set on potential druggable pockets identified on the protein surface. Then, virtual screening campaigns were run using Glide (version 6.1) from Schrödinger as the docking software[49], and a database of lead-like and commercially available compounds

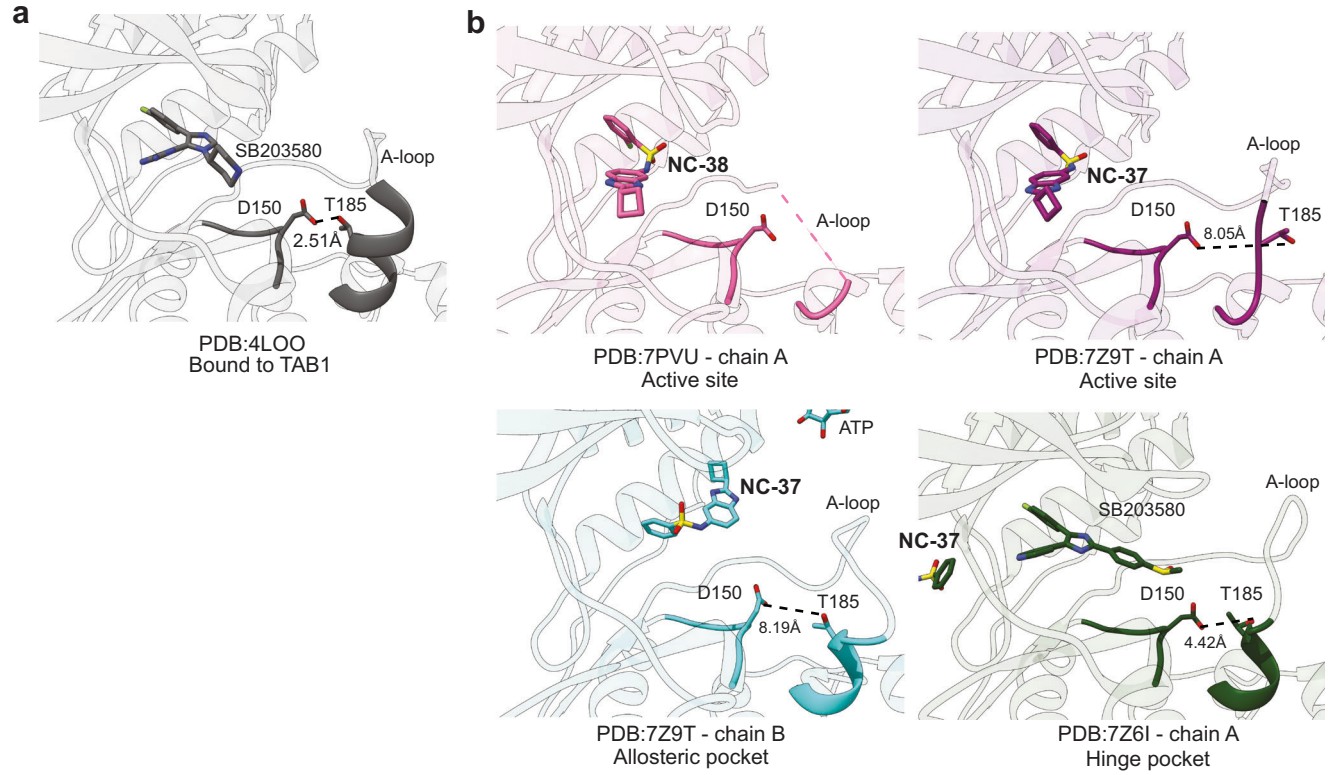

**Fig. 9 | NC-p38i compounds disrupt the formation of a hydrogen bond between Thr185 and Asp150 in p38α. a** Crystal structure of the p38α-TAB1 complex showing the hydrogen bond (2.5 Å) formed between Thr185 and Asp150 that is required for TAB1-induced autophosphorylation of p38α[7]. **b** Crystal structures of

the three p38α-NC-p38i complexes presented in this manuscript showing that Thr185 and Asp150 are either too far apart or not properly positioned for the formation of the critical hydrogen bond. Distances are indicated in Å.

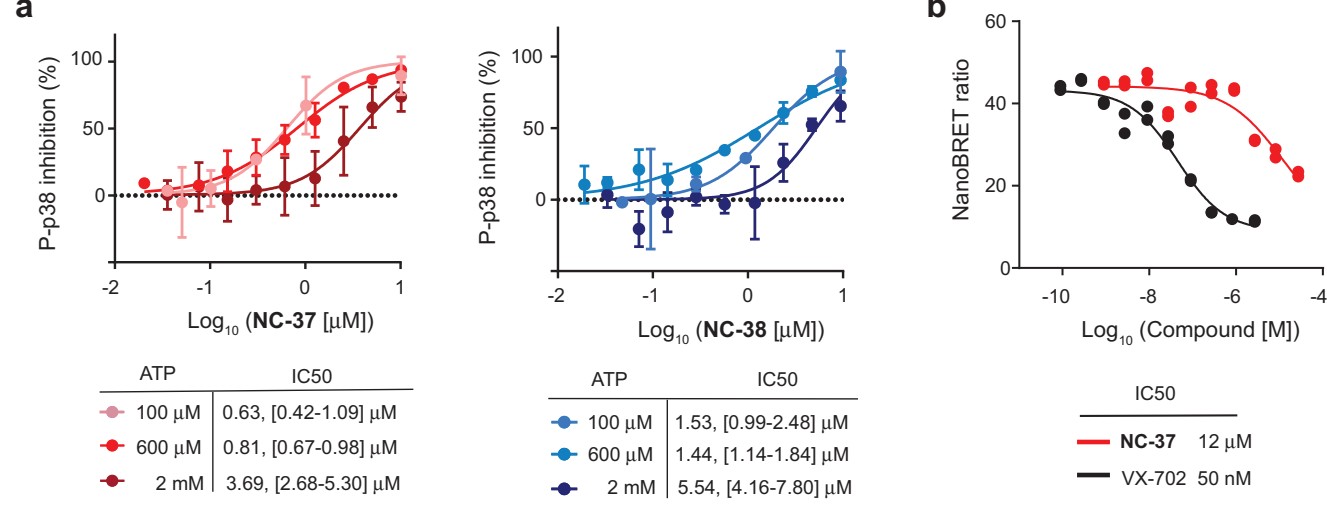

**Fig. 10 | NC-p38i compounds behave as weak ATP-competitive inhibitors.**
**a** Inhibition curves of p38α autophosphorylation by different concentrations of compounds **NC-37** and **NC-38** (0.035 to 10 μM) at the indicated ATP concentrations. Results are shown as mean ± SD (*n* = 3 experiments). Data were fitted using a nonlinear regression fit model (Graphpad Prism) to determine the IC50s, with 95% confidence intervals indicated in brackets. **b** HEK293T cells expressing NanoLuc-fused p38α were incubated with a cell-permeable energy transfer probe, which

generates a BRET signal upon binding to the ATP site of p38α. Cells were incubated with different concentrations of compound **NC-37** or the ATP-competitive inhibitor VX-702 as a control, and the NanoBRET ratio was measured to calculate the IC50. Results are shown for two technical repeats from a representative experiment. In total two biologically independent experiments were performed and gave similar results. Source data are provided as a Source data file.

extracted from the LEADS-LIKE NOW library included in the ZINC12 database (2,419,472 compounds as of 1 November 2013)[26]. The initial subset was expanded to include tautomeric, stereochemical, and ionization variations using LigPrep from Schrödinger [LigPrep version

2.8, Schrödinger, LCC; New York, NY, 2013]. The top 200-ranked compounds in terms of Glide Score were visually inspected and those with better physicochemical properties and interactions with the receptor were selected. The final set of 100 compounds referred to as

Virtual Screening 1 (VS1) included 4 predominant scaffolds. Then, a substructure similarity search based on these scaffolds was performed in the ZINC12 database to extend the selection to compounds with molecular weight >350 and at least 90% similarity to the mentioned scaffolds. This exercise allowed us to find 5548 similar compounds that were subjected to docking with Glide, and 51 new structures referred to as VS2 were obtained and added to the VS1 set. Further ICM clustering by structure[50] to select the centroids with better GlideScore gave a total of 50 compounds. These were again visually inspected to choose the final set of 35 compounds that included hit **NC-27** and were experimentally validated (Supplementary Fig. 6).

A substructure similarity search was performed on the whole ZINC database using as reference the scaffolds of selected compounds including **NC-27**. Compounds with >90% similarity to the references were subjected to similar docking studies and the most promising ones in terms of docking score, binding mode diversity and commercial availability were selected. In parallel, the FDA library of approved compounds was screened using the same grids, and the LEADS-LIKE-NOW library was also screened using an average structure extracted from a Molecular Dynamics of p38α (PDB:4LOO). In total, 52 additional compounds were purchased from commercial databases and other 46 compounds were synthesized, which were all tested in the p38α autophosphorylation assay.

The commercial suppliers of compounds are indicated in Supplementary Table 1.

## Protein expression and purification
The plasmids used for recombinant protein expression are indicated in Supplementary Table 2.

**GST-tagged proteins.** Human p38α and p38β proteins with an N-terminal GST-tag were produced in *E. coli* BL21 DE3 or BL21-DE3-pLysS. Bacteria were grown in 500 ml of LB medium containing ampicillin (50 µg/ml) at 37 °C until an $OD_{600}$ of 0.6–0.8 and induced for 3 h at 18 °C with 0.05 mM IPTG. GST-MK2, GST-ATF2, and GST alone were induced for 3 h at RT with 1 mM IPTG.

After induction, cells were harvested by centrifugation ($3220 \times g$ for 10 min at 4 °C), resuspended in 13.5 ml of cold PBS containing 1 mg/ml lysozyme, 5 mM EDTA, 1 complete EDTA-free protease inhibitor cocktail tablet, and sonicated. Then, Triton X-100 was added to a final concentration of 1% (v/v), and cell debris were cleared by centrifugation ($9400 \times g$ for 20 min at 4 °C). Glutathione-Sepharose 4 Fast Flow beads were washed in cold PBS buffer, resuspended in PBS (50:50), and 70 µl of the bead slurry were added per 1 ml of supernatant. After 2 h rotating at 4 °C, samples were centrifuged ($200 \times g$ for 2 min at 4 °C) and the supernatant was removed. Beads were washed three times with 10 ml of cold PBS and once with 10 ml of cold 50 mM Tris-HCl pH 8.0, and then centrifuged to remove residual buffer. Beads were incubated with 1 ml of 10 mM L-glutathione in 50 mM Tris-HCl pH 8.0 and protein elution was performed at RT. To remove glutathione, eluted proteins were put into a dialysis bag (12–14 MWCO) and dialyzed against 2 l of buffer containing 20 mM Tris pH 8.0, 50 mM NaCl, 0.1 mM EDTA, 0.5 mM DTT, and 5% glycerol, overnight at 4 °C. Following dialysis, samples were collected, aliquoted, and stored at −80 °C. An aliquot was analyzed in 10% SDS-PAGE and stained with Coomassie blue to quantify the purified proteins using a BSA standard curve as a reference.

**His-SUMO-p38α protein.** The construct to express in *E. coli* the mouse p38α C162S mutant with an N-terminal His-tag followed by a SUMO protease cleavage sequence was generated using Gibson cloning strategy protocols[51].

Rosetta-DE3-pLysS pre-cultures were grown in LB medium containing ampicillin (50 µg/ml) at 25 °C until an $OD_{600}$ of 0.5 and were induced for 3 h at 18 °C with 0.2 mM IPTG. Alternatively, BL21 cells were grown at 37 °C to reach an $OD_{600}$ of 0.8–1.0 and induced overnight at 20 °C with 0.2 mM IPTG. After induction, cells were harvested by centrifugation (3500 g for 10 min at 4 °C) and pellets were stored at −80 °C until protein purification.

Cells from 1 l of culture were thawed and resuspended in 50 ml of cold lysis buffer containing 50 mM Tris pH 7.4, 500 mM NaCl, 10 mM imidazole and 1 complete EDTA-free protease inhibitor cocktail tablet, and sonicated. Cell debris were cleared by centrifugation ($15,900 \times g$ for 30 min at 4 °C) and the supernatant was passed through a 0.45 µm filter. Then, p38α was purified by FPLC using an NGC Quest 10 Plus Chromatography system (BioRad) on a HisTrap HP column 5 ml (GE Healthcare, 17-5248-01) at 4 °C, which was previously equilibrated with buffer A (50 mM Tris pH 7.4, 500 mM NaCl, 10 mM imidazole). The protein of interest was eluted by applying 16 CV (column volumes) of a linear 10 mM to 500 mM gradient of imidazole.

Fractions from each chromatographic step were analyzed by 12% SDS-PAGE and stained with Coomassie blue. Cleavage of His-SUMO-tag was performed overnight at 4 °C (6–8 kDa membrane) with in-house purified recombinant His-Tagged SUMO protease (50 µl at 2.5 mg/ml for 2.5 mg of fusion protein) against 3 l of dialysis buffer containing 25 mM Tris pH 7.4, 150 mM NaCl, 10 mM $MgCl_2$ and 1 mM DTT. Cleaved p38α was checked by SDS-PAGE and stained with Coomassie blue. The cleaved protein was diluted to 50 mM of NaCl and was loaded onto a HiTrap Q HP anion exchange column of 5 ml (GE Healthcare, 17-1154-01) at 4 °C, which was previously equilibrated with buffer A (50 mM Tris pH 7.4, 50 mM NaCl, 10 mM $MgCl_2$, 5% glycerol and 1 mM DTT) and buffer B (50 mM Tris pH 7.4, 500 mM NaCl, 10 mM $MgCl_2$, 5% glycerol and 1 mM DTT) (5 CV of buffer A + 5 CV of buffer B + 5 CV of buffer A). The protein was eluted by applying 40 CV of a linear 50 mM to 500 M NaCl gradient. Fractions from each chromatographic step were analyzed by 12% SDS-PAGE and stained with Coomassie blue. Fractions containing p38α were further purified by size-exclusion chromatography using a prepgrade HiLoad 16/600 Superdex 75 column (GE Healthcare) in buffer C (20 mM Tris pH 7.5, 100 mM NaCl, 10 mM $MgCl_2$, and 5 mM TCEP). Pure p38α protein was quantified using a NanoDrop One Microvolume UV-Vis Spectrophotometer (Thermo Scientific), then was concentrated to 10–15 mg/ml with Amicon Ultra-4 filter units (10 kDa) and used for crystallography. Protein integrity and phosphorylation were confirmed by immunoblotting.

## TAB1-induced p38α autophosphorylation
A peptide corresponding to amino acids 386–414 of human TAB1 [RVYPVSVPYSSAQSTSKTSVTLSLVMPSQ] (GenScript) was resuspended in 50 mM Tris-HCl pH 7.5, at a concentration of 100–300 mM. Non-phosphorylated GST-p38α protein (2 µg) was incubated with 15 µM TAB1 peptide and 600 µM ATP in 20 µl of autophosphorylation buffer containing 100 mM NaCl, 20 mM Tris-HCl pH 7.5, 2 mM DTT, and 2 mM $MgCl_2$, for 2 h at 37 °C. Reactions were stopped by the addition of 5X sample loading buffer and boiling at 95 °C for 5 min. Autophosphorylation was detected by immunoblotting using antibodies that recognize the two phosphorylation sites in the activation loop of p38α.

## Basal p38α and p38β autophosphorylation
Non-phosphorylated GST-p38α and GST-p38β proteins (2 µg) were incubated with 600 µM ATP in 20 µl of autophosphorylation buffer for 2 h at 37 °C. Reactions were stopped by the addition of 5X sample loading buffer and boiling at 95 °C for 5 min. Autophosphorylation was detected by immunoblotting as above.

## Phosphorylation of p38α by MKK6
Non-phosphorylated GST-p38α protein was incubated with purified MBP-MKK6$^{DD}$ protein (4:1) in 20 µl of kinase buffer containing 50 mM Tris-HCl pH 7.5, 2 mM DTT, 10 mM $MgCl_2$, 100 µM $Na_3VO_4$, 1 mM PMSF, 10 µg/ml aprotinin, 10 µg/ml leupeptin and 200 µM ATP. After 30 min

at 37 °C, reactions were frozen to be used for substrate phosphorylation (see below) or were mixed with 5X sample loading buffer, boiled at 95 °C for 5 min and analyzed by immunoblotting using antibodies that recognize the two phosphorylation sites in the p38α activation loop.

## Phosphorylation of substrates by active p38α

MKK6$^{DD}$-actived GST-p38α was incubated with purified GST-ATF2 or GST-MK2 proteins (1:4) in 20 μl of kinase buffer with ATP for 30 min at 30 °C. Reactions were stopped by the addition of 5X sample loading buffer and boiling at 95 °C for 5 min. Phosphorylation of ATF2 or MK2 was detected by immunoblotting using antibodies that recognize the corresponding p38α phosphorylation sites.

## Immunoblotting

Cells were lysed in 1x RIPA buffer (50 mM Tris-HCl, 150 mM NaCl, 1% NP-40, 5 mM EDTA, 1 mM DTT, 1 mM Na$_3$VO$_4$, 1 mM PMSF, 10 μg/ml pepstatin A, 10 μg/ml aprotinin, 10 μg/ml leupeptin, 20 mM NaF, 1 μM microcystin, 2.5 mM benzamidine) and protein concentrations was determined using the RC DC protein assay kit (BioRad) according to manufacturer's instructions. Proteins were separated by 8–10% SDS-PAGE and transferred to nitrocellulose membranes using a wet-transfer system (Mini Trans-Blot Cell, BioRad). Membranes were stained with Ponceau S solution and blocked in 5% non-fat dry milk in 1X PBS for 1 h at RT. Primary antibodies were diluted in 5% BSA in TBS-Tween 0.05% and incubated overnight at 4 °C or for 2 h at RT. Then, membranes were washed in 1X PBS and incubated with the secondary antibody diluted in TBS-Tween 0.05% with 2.5% BSA for 1 h at RT. Finally, membranes were washed three times as above and protein-bound antibodies were detected using the LI-COR Odyssey system. The antibodies used are indicated in Supplementary Table 3.

## Thermal shift assays

For purified proteins, samples were prepared in triplicates by mixing in MultiPlate 96-well white plates purified GST-p38α or GST proteins (1.5 μM) with assay buffer (20 mM TRIS pH 7.5, 100 mM NaCl, 2 mM MgCl$_2$, 2 mM DTT), Protein Thermal Shift buffer (1X final concentration, Applied Biosystem Protein Thermal Shift™ Kit) and NC-p38i compounds at different concentrations (10 mM stock in 100% DMSO) or the equivalent volume of DMSO as a control. After 30 min incubation at RT, 1X SYPRO Orange was added and fluorescence intensity was measured at 1 °C intervals from 25 to 95 °C at a rate of 0.5 °C/s (CFX96, BioRad). All measurements were performed in triplicates and mean values were reported for all compounds to calculate their ΔTm relative to the control DMSO samples.

For intact cells, KBM7 cells were treated with DMSO or NC-p38i for 2 h at 37 °C ($2 \times 10^6$ cells/ml). Then, cells were centrifuged and divided into PCR tubes ($10^6$ cells/tube). PCR tubes containing pellets of treated cell were heated for exactly 3 min at the indicated temperatures. Subsequently, 30 μl of buffer (20 mM Tris-HCl pH 8, 150 mM NaCl and 0.5% NP-40) were added to each tube and cells were lysed by freeze-thaw cycles. Then, lysates were centrifuged ($15,900 \times g$ for 30 min at 4 °C) and supernatants (soluble fraction) were analyzed by immunoblotting.

## Fluorescence polarization

Assays were performed in low volume 384-well plates (non-binding surface, black, Corning, 3575) using an Infinite® 200 (Tecan) microplate reader. GST-p38α protein was dialyzed overnight in a buffer containing 20 mM Tris pH 7.5, 100 mM NaCl, 1 mM DTT, and 10 mM MgCl$_2$. The stock of fluorescent N-terminally-labeled FITC-TAB1$_{386-414}$ peptide (ChinaPeptides) was prepared at 10 mM in DMSO.

For direct binding assays, FITC-TAB1$_{386-414}$ (10 nM) and non-phosphorylated GST-p38α (0.17–22 μM) were prepared in triplicates in the above-mentioned buffer and incubated for 1 h at RT. The FP signal (mP) was measured at an excitation wavelength (λex) of 485 nM and

emission wavelength (λem) of 535 nM. The concentration of GST-p38α that increased the FP signal up 80% was used to perform the competitive assays.

For competition assays, non-phosphorylated GST-p38α (5.6 μM) was prepared in triplicates in the above buffer containing increasing concentrations of NC-p38i (diluted from a 10 mM stock in 100% DMSO). The equivalent volume of DMSO (1%) was added as a control. The FP signal was measured after incubation for 1 h at RT.

## Crystallography

High-throughput crystallization screening and optimization experiments were performed at the Automated Crystallography Platform (PAC) of IBMB-CSIC and IRB Barcelona.

Full-length mouse p38α C162S mutant protein (10–15 mg/ml) was crystallized in presence of different combinations of NC-p38i (p38α:NC-p38i, 1:1.5 ratio), SB203580 (1 mM), and ATPγS (500 μM) in a solution containing 20 mM Tris pH 7.5, 100 mM NaCl, 10 mM MgCl$_2$, and 5 mM TCEP as previously described[52]. Standard screenings and optimizations were prepared with a Tecan Evo 100. Experiments were dispensed in total drop volumes of 200 nl each (1:1 ratio) using a Phoenix protein dispenser from ARI. Crystal complexes were grown by sitting-drop vapor diffusion and monitored on Bruker Crystal Farms. Crystal growth conditions of the 3 presented structures are listed in Supplementary Table 4.

The X-ray data were collected at 100 K from monoclinic and orthorhombic crystals using a PILATUS 6 M detector on BL13-XALOC at the ALBA Synchrotron Light Source (Barcelona)[53]. Further details are indicated in Supplementary Table 4.

Data reduction and processing were either carried out using imosflm[54], scala[55], and ctruncate[56] from the CCP4 suite[57], or the autoPROC pipeline implemented at the Alba synchrotron[58]. Besides, Phenix software suite was also used[59]. Structures were solved by molecular replacement with phaser[60] using as a reference the pdb (www.wwpdb.org) structure 4LOO [https://doi.org/10.2210/pdb4LOO/pdb]. Refinement was carried out using either Phenix.refine[61] or refmac5[62] and manually with Coot[63].

Polder maps were created for every ligand to ensure the correct placement of its atoms. Structure 7Z6I [https://doi.org/10.2210/pdb7Z6I/pdb] was verified and ligand's presence and position confirmed (Supplementary Fig. 7a). Furthermore, a series of polder maps were used to successfully build loop 170–182, whose signal was not well visible in other presented structures. The occupancy values of these atoms were set to zero. In the case of 7Z9T [https://doi.org/10.2210/pdb7Z9T/pdb] both chain A (Supplementary Fig. 7b) and B of NC-37 ligand were confirmed to be present, however, we were not able to unequivocally eliminate the alternative orientation of the chain B ligand, thus its occupancy was set to 0. For 7PVU [https://doi.org/10.2210/pdb7PVU/pdb] (ligand NC-38) the electron density in the chain A suggested a presence of the compound, while polder maps allowed for pinpointing the compound's orientation before subsequent refinement steps (Supplementary Fig. 7c). No credible compound signal could have been identified in the chain B. Polder maps were calculated using phenix.polder[64] without searched atoms in place and exclusion sphere radius within 5–10 Å, to minimize bias. The correctness of masks was also verified. Validation of polder maps was performed following the approach described in ref. 64. All ligands reported using polder maps in this paper have CC(1,3) values larger than CC(1,2) and CC(2,3), which is the acceptance criterion.

We verified that no other densities could be attributed to the ligands using Coot[63]. The "Unmodeled blobs" Coot option did not report any unmodeled blobs in the 2mFo-DFc map for blobs even above the RMSD of 1.0. Finally, in regions where water appeared to form chains, we generated polder maps to see if they could be ligands in place. This procedure revealed no additional ligands and confirmed that the density belonged to water molecules.

All graphical representations were prepared using either Chimera[65] or Coot.

## Cell maintenance

HEK293T (ATCC, CRL-3216), H9c2 (ATCC, CRL-1446), U2OS (ATCC, HTB-96), and BBL358[66] cells were cultured in DMEM (high glucose) medium supplemented with 10% FBS, 1% penicillin/streptomycin and 1% L-glutamine. KBM7 cells (Horizon Discovery, C628) were cultured in IMDM medium supplemented with 10% FBS and 1% penicillin/streptomycin. Cells were maintained at 37 °C and 5% $CO_2$.

## Cell transfection

HEK293T cells were transiently transfected at 60–70% confluency. Mouse myc-p38α and GFP-TAB1 in the pcDNA3 vector were transfected using $CaCl_2$. Briefly, DNA (5 μg myc-p38α, 10 μg GFP-TAB1) was dissolved in 450 μl of sterile water with 50 μl of 2.5 M $CaCl_2$ and incubated for 5 min at RT. Afterward, 500 μl of 2X HBS were added and incubated for 20–30 min at RT. The transfection mix was added to cells and incubated overnight. Next day, the media was replaced and cells were seeded to perform experiments after 24 h.

## TAB1 knockdown in H9c2 cells

H9c2 cells were transiently transfected at 70–80% confluence with a TAB1 siRNA (Thermo Fisher, cat # AM51331) using Lipofectamine RNAiMAX reagent (Invitrogen). Briefly, 9 μl of Lipofectamine and 1.2 μl of TAB1 siRNA (25 μM) were each dissolved in 150 μl of Opti-MEM. The diluted siRNA was mixed with the diluted Lipofectamine (1:1 ratio) and incubated for 5 min at RT. Afterward, 250 μl of the transfection mix was added to 500,000 cells in 2 ml of Opti-MEM (6-well plate) and incubated for 7 h. The media was then replaced for complete DMEM and cells were incubated for 72 h prior to the experiments.

## Simulated ischemia-reperfusion (SIR) in H9c2 cells

H9c2 cells were seeded in 60 mm plates and grown for 24 h. Then, medium was replaced and NC-p38i compounds or ATP-competitive inhibitors of p38α were added for a 24 h pre-treatment. For simulated ischemia, the medium was changed to ischemic buffer (137 mM NaCl, 12 mM KCl, 0.5 mM $MgCl_2$, 0.9 mM $CaCl_2$, 4 mM HEPES, 10 mM 2-deoxy-glucose and 20 mM sodium lactate, pH 6.2)[67] containing either DMSO, ATP-competitive inhibitors or NC-p38i compounds. Cells were kept in a H35-hypoxistation chamber flushed with 0.1% $O_2$, 5% $CO_2$, and 9% $N_2$ (65% HR) for 2 h. For reperfusion, cells were switched back to the original medium and kept for 15 min or 4 h in a normal incubator with 5% $CO_2$ (normoxia). Non-treated cells were maintained in a normal incubator with 5% $CO_2$.

After reperfusion, the media containing dead cells as well as the washes with 1x DPBS were collected. Cells were trypsinized, added to the collected media, and the suspension was centrifuged at $450 \times g$ for 5 min. Then, cells were either stained with Annexin V/PI and analyzed by flow cytometry, or lysed in 1X RIPA buffer and the supernatants were analyzed by immunoblotting.

## Analysis of H9c2 cell death by Annexin V/PI staining

The medium containing dead cells was collected, and cells were washed with 1X DPBS, which was also collected. Then, cells were trypsinized and the suspension was combined with the medium and the 1X DPBS washing, and centrifuged at $450 \times g$ for 5 min. The supernatant was removed and the cell pellet was resuspended in 1X Binding Buffer (FITC Annexin V Apoptosis Detection Kit I, BD Pharmingen) to yield a $1 \times 10^6$ cells/ml suspension. Next, 5 μl of FITC Annexin V were added to 100 μl of the cell suspension and incubated for 20 min at RT under dark conditions. The resulting mixture was diluted in 300 μl of 1X Binding Buffer and 5 μl of PI (50 μg/mL) were added. Cell death was analyzed in a Gallios Flow Cytometer (Beckman

Coulter) using the FlowJo software. FACS gating strategy is shown in Supplementary Fig. 8.

## NanoBRET target engagement assay

HEK293T cells were resuspended in Opti-MEM containing 1% FBS and transfected in suspension with a mix containing Transfection Carrier DNA (9 μg/ml), NanoLuc fusion p38α vector (1 μg/ml) and 30 μl of FuGENE HD in Opti-MEM without serum. Then, cells were seeded into 96-well plates for 20 h to allow the expression of NanoLuc-fused p38α.

The NanoBRET Tracer (100X in DMSO) was diluted 1:4 in Tracer Dilution Buffer (20x) and added directly to cells (1:20), which were then mixed for 15 s at $100 \times g$. NC-p38i compounds were prepared by serial dilutions in Opti-MEM and added to cells. Plates were again mixed for 15 s at $100 \times g$ and further incubated for 2 h at 37 °C 5% $CO_2$. Finally, plates were incubated for 15 min at RT to cool down before performing BRET measurements.

Immediately prior to BRET measurements, 3X Complete Nano-BRET Nano-Glo Substrate was prepared in OptiMEM without serum or phenol red (1:166 dilution of NanoBRET Nano-Glo Substrate plus a 1:500 dilution of Extracellular NanoLuc Inhibitor in OptiMEM without serum or phenol red) and mixed gently by inversion. 50 μl were added per well of 3x Complete NanoBRET Nano-Glo Substrate and incubated 2–3 min at RT. Measurements were done within 10 min of substrate activation. Following addition of NanoBRET Nano-Glo Substrate, donor emission (e.g., 450 nm) and acceptor emission (e.g., 610 nm or 630 nm) were measured using a NanoBRET-compatible luminometer. To generate raw BRET ratio values, acceptor emission values (e.g., 610 nm) were divided by the donor 73 emission value (e.g., 450 nm) for each sample. Then, raw BRET units were converted to % inhibition for interpretation of the results. These experiments were performed by Aurelia Biosciences.

## Kinase selectivity panels

NC-p38i compounds were tested by ThermoFisher in a panel of 97 human kinases using the Z'-LYTE technology, and by Eurofins using the KINOMEscan™ Profiling Service in a panel of 468 human protein kinases.

## Statistical analysis and reproducibility

Data are expressed as mean or mean ± SD, and statistical analysis was performed by using a two-tailed Student's t-test for the comparison of two groups using GraphPad Prism Software 9.0.0 (GraphPad Software, Inc., La Jolla, CA), unless otherwise stated in the figure legends. $p$-values are expressed as follows: $^*p \le 0.05$, $^{**}p \le 0.01$, $^{***}p \le 0.001$ and $^{****}p \le 0.0001$. Immunoblots were repeated at least three independent times unless stated otherwise in the figure legends.

## Data availability

The diffraction data and coordinates of the p38α complexes bound to NC-p38i compounds have been deposited in the Protein Data Bank under accession codes 7PVU, 7Z6I and 7Z9T. We have also used the following PDB structures: 4LOO, 1A9U, 3COI, 7N8T, 2ZOQ, 1PME, 3GC9, 1CM8, 4UX9. Source data are provided with this paper.

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

## Acknowledgements

We thank Xandra Kreplin and Roman Bonet from the Institute of Molecular Biology of Barcelona (IBMB)-CSIC for performing early crystallization trials. We also thank the staff of the Mass Spectrometry Core Facility (Universitat de Barcelona) for support, the Protein Expression Core Facility (IRB) for providing reagents, and the Automated Crystallography Platform staff (IRB-CSIC) and ALBA Synchrotron staff for support with the experiments performed at the BL13-XALOC beamline, and Xavier Salvatella, Tiago Oliveira and Israel Ramos from IRB for support and discussions. This work was supported by grants from the Spanish Ministerio de Ciencia e Innovación (MICINN, PID2019-109521RB-I00 and PID2021-122478NB-I00), the BioMedTec program of IRB-Fundació La Caixa, the European Research Council (Proof of Concept p38_InTh-825763), AGAUR (2016 LLAV 00043 and 2019 PROD 00138 supported by FEDER, and 2017 SGR-557, 2017 SGR-50, 2021 SGR-909, and 2021 SGR-866), BBVA Foundation, and the European Union's Horizon 2020 research and innovation program (euCanSHare 825903 and BioExcel-3 101093290). L.G. and B.B. were funded by predoctoral contracts from MICINN (BES-2016-077122) and the Marie Skłodowska-Curie COFUND action of IRB Barcelona and the PREBIST Predoc Programme (PREBIST_754558), respectively. F.C. is a Ramon y Cajal Fellow (RYC2019-026768-I). Access to ALBA was granted through the BAG proposals 2018092972 and 2020094472. We gratefully acknowledge institutional funding from IRB Barcelona, the CERCA Programme of the Catalan Government, and the MICINN through the Centres of Excellence Severo Ochoa award. M.J.M. and A.R.N. are supported by the Institució Catalana de Recerca i Estudis Avancats (ICREA).

## Author contributions

L.G., A.D.-C., M.S., I.B.-H., A.I., and L.R. performed biochemical, biophysical, and in cells experiments and analyzed the data. L.D., Ch.P., M.E., F.C., and R.S. designed and performed in silico screenings and modeling. J.P., B.B., P.M., and M.J.M. performed crystallography studies and determine the structures. C.M.-R. and R.M.B. provide essential reagents and advice. L.G., A.D.-C., M.J.M., M.O., and A.R.N. wrote the manuscript with input from all authors. M.J.M., M.O., and A.R.N. supervised the study and were involved in the experimental design and data interpretation.

## Competing interests

IRB Barcelona, UB, ICREA, BSC-CNS, and Nostrum Biodiscovery have filed the patent application WO2020120576 - P38A AUTOPHOSPHORYLATION INHIBITORS. L.G., L.D., A.I., R.S., M.O., and A.R.N. are named inventors on this application, but the patent was abandoned. The rest of the authors declare no competing interests.
