## [Peer review file · Nature Communications]

REVIEWER COMMENTS

Reviewer #1 (Remarks to the Author):

Manuscript named: "Characterization of p38a autophosphorylation inhibitors that target the non-canonical

activation pathway" presented very interesting study of novel type of inhibitors of p38a.

Results are impressive with providing full experimental confirmations for inhibition of SIR-triggered cell death.

My opinion is that the manuscript is suitable for publication in Nature Communications journal.

Reviewer #2 (Remarks to the Author):

Authors presented the outcome of a small molecule screening campaign focusing on TAB1 mediated non-canonical p38 activity. Using extensive biochemical methods, the inhibitory characteristics of the hit-compounds were analyzed. The functional effects were demonstrated in cultured cells. The inhibitory mechanism was detailed based on interaction and co-crystal analysis. These data showed several interaction pockets in p38a for the lead compounds, and offered potential structural basis for the inhibitory effects and specificity.

The discovery of novel class of p38 inhibitors targeted towards its non-canonical activation mechanism can be potentially important from both conceptual understanding and pharmacological application. However there are several issues regarding the conclusion and significance of these findings.

1). It is not clear if the top two hits are derivatives from the #27 compound and why it is not included in the functional assay against I/R induced cell death? A flowchart of the screening campaign would be helpful.

2). While the screening was based on docking sites for TAB1-p38 interaction and inhibitions towards TAB1 mediated p38a autophosphorylation, the biochemical results showed the two top hits appeared to have no effects on TAB1-p38a interaction, but rather had a weak interaction to the ATP binding pocket, in addition to other two areas. How does the binding to each pocket contribute to overall inhibitory

effect (individually or synergistically) remains unclear. Therefore, the structural basis for the specificity in their inhibitory function on TAB1 mediated p38 autophosphorylation needs to be better demonstrated. It is not clear if the compounds can bind to different sites simultaneously or exclusively? There was no site-directed mutagenesis data to demonstrate the functional impact of the different binding pockets for p38 inhibition.

3). The functional effects against p38a activation was tested in H9c2 cardiomyocytes following hypoxia/reoxygenation injury and other classic stressors. Authors should present evidence of p38 alpha activities from the experimental samples following I/R injury in correlation with caspase cleavage activities. Direct evidence of apoptosis should be demonstrated (TUNEL) and dependence on TAB1 specific p38 activation should be supported by TAB1 KD in the cells in combination with inhibitor treatment.

Reviewer #3 (Remarks to the Author):

The authors describe the identification of a new class of ligands they suggest is able to block p38a autophosphorylation but poorly affects the activity of the canonical pathway. The discovery of p38 pathway-specific inhibitors has the potential to be of high impact, substantial evidence in the medical literature shows that blanket pharmacological inhibition of p38 activity, by ATP-competitive inhibitors causes adverse side effects most likely due to on target effects; the development of inhibitors that are circumstance specific could help overcome the side effects observed with blanket inhibition.

The manuscript is well written, and the experiments are carried out to a high standard. A large number of experiments is presented both in cells and in vitro, but overall, in my opinion, they do not constitute sufficient evidence to support the conclusion that the ligands are selective in inhibiting p38 signalling through autophosphorylation. The difference they observe between transphosphorylation and autophosphorylation inhibition could be simply down to the fact that the enzyme substrate ratio will be significantly different between the two types of activation; in the autophosphorylation reaction p38 is both substrate and enzyme whereas in the transphosphorylation reaction the substrate is in large excess compared to the enzyme. For instance, it is not clear why they use molarity unit- measures when reporting the concentration of the reagents in the autophosphorylation reaction and instead weight/volume when reporting the transphosphorylation reaction. A major issue might be that in the transphosphorylation reaction they are operating at a kinase concentration that is too low for the

inhibitors to interact, therefore making the comparison not reliable. At the moment given the different unit measures used it is not possible to compare.

The structural data do not provide sufficient insight into the mechanism through which the ligands might specifically inhibit autophosphorylation. An explanation, at a molecular level, of the selectivity of the ligands might help support their hypothesis. I would suggest the authors check whether ligand binding affects the formation the h-bond between Thr185 and Asp150, which is crucial for the autophosphorylation reaction; mutations that affect the h-bond formation selectively abolish autophosphorylation (PDB:5O90).

It also not clear what is the binding stoichiometry of the interaction between the ligands and the kinase. The authors write that the structures they solved reveal that NC-p38i compounds predominantly occupy three different areas around the active cavity. What do they mean by predominantly? Can they please report whether they observe any extra unassigned electron density in their maps that could be attributed to the ligands.

The structures of the NC-p38i compounds in complex with p38 were obtained through co-crystallization experiments. Based on the 1:1.5 ratio they report, the protein concentration is between 0.2-0.4mM whereas the NC-p38i compounds are at 0.3-0.6mM, whereas for SB203580 (a much stronger binder) they use a larger excess (1mM), what is the rationale for their choice. Moreover the authors should comment on whether the crystal packing has any effect on the NC-p38i compounds binding mode they observe.

In the protein expression section of the manuscript, it is described how the cells expressing His-SUMO-p38a protein are grown in deuterated water and minimal media, what is the reason for this?

Finally the authors should report the CC values for the polder maps in Fig. S6.

Manuscript NCOMMS-22-42583-T. Response to Reviewers.

Reviewer #1:

Manuscript named: "Characterization of p38a autophosphorylation inhibitors that target the non-canonical activation pathway" presented a very interesting study of novel type of inhibitors of p38a.

Results are impressive with providing full experimental confirmations for inhibition of SIR-triggered cell death.

My opinion is that the manuscript is suitable for publication in the Nature Communications journal.

Thank you.

Reviewer #2:

Authors presented the outcome of a small molecule screening campaign focusing on TAB1 mediated non-canonical p38 activity. Using extensive biochemical methods, the inhibitory characteristics of the hit-compounds were analyzed. The functional effects were demonstrated in cultured cells. The inhibitory mechanism was detailed based on interaction and co-crystal analysis. These data showed several interaction pockets in p38a for the lead compounds, and offered potential structural basis for the inhibitory effects and specificity.

The discovery of a novel class of p38 inhibitors targeted towards its non-canonical activation mechanism can be potentially important from both conceptual understanding and pharmacological application. However there are several issues regarding the conclusion and significance of these findings.

1. It is not clear if the top two hits are derivatives from the #27 compound and why it is not included in the functional assay against I/R induced cell death? A flowchart of the screening campaign would be helpful.

We have included a flowchart of the screening campaign that led to the identification of the hit compound #27 indicating the number of molecules at each stage (new Supplementary Fig. 6), and also added further information on the screening in Methods. Moreover, we have stated in the text that compounds #37 and #38 are derivatives of #27, as shown in Supplementary Fig. 1. We chose compounds #37 and #38 for the functional assays because they showed higher potency than #27 in the *in vitro* autophosphorylation assays.

2. While the screening was based on docking sites for TAB1-p38 interaction and inhibitions towards TAB1 mediated p38a autophosphorylation, the biochemical results showed the two top hits appeared to have no effects on TAB1-p38a interaction, but rather had a weak interaction to the ATP binding pocket, in addition to other two areas. How does the binding to each pocket contribute to overall inhibitory effect (individually or synergistically) remains unclear. Therefore, the structural basis for the specificity in their inhibitory function on TAB1 mediated p38 autophosphorylation needs to be better demonstrated. It is not clear if the compounds can bind to different sites simultaneously or exclusively? There was no site-directed mutagenesis data to demonstrate the functional impact of the different binding pockets for p38 inhibition.

We did not observe simultaneous binding of multiple compounds in any of the solved structures, and our hypothesis is that a NC-p38i compound mainly binds a specific single site located in and around the catalytic site. However, we cannot exclude the possibility that one

molecule could bind at the hinge and another one at the catalytic site simultaneously in solution, despite not having detected this situation in any of our crystal structures.

Regarding the contribution of each pocket to the inhibitory effect, we have not been able to observe in any of the structures of p38 α with NC-p38i compounds the hydrogen bond between the Thr185 and Asp150 side chains (new Fig. 7), which was previously described to be required for TAB1-mediated p38 α autophosphorylation¹. These observations suggest that binding to any of the three pockets inhibits p38 α autophosphorylation by preventing formation of the hydrogen bond. Please see further comments below in the reply to Query 2 of Reviewer #3.

Given the position of the identified binding pockets, it is very likely that site-directed mutagenesis of these residues would kill the kinase activity of p38 α precluding to perform the autophosphorylation assays needed to test the relevance of the mutated sites.

3. The functional effects against p38a activation was tested in H9c2 cardiomyocytes following hypoxia/reoxygenation injury and other classic stressors. Authors should present evidence of p38 alpha activities from the experimental samples following I/R injury in correlation with caspase cleavage activities. Direct evidence of apoptosis should be demonstrated (TUNEL) and dependence on TAB1 specific p38 activation should be supported by TAB1 KD in the cells in combination with inhibitor treatment.

To address this point, we have evaluated the effects of NC-p38i in cell death triggered by simulated ischemia/reperfusion (SIR). Using Annexin V and propidium iodide labelling followed by flow cytometry analysis, we observed a significant reduction in SIR-induced death of H9c2 cells upon treatment with either NC-p38i or the ATP-competitive inhibitor SB203580, in comparison with the untreated controls (new Fig. 1e). In addition, we confirmed that siRNA-mediated downregulation of TAB1 significantly reduced SIR-induced death of H9c2 cells (new Fig. 1f), in agreement with the results obtained upon treatment with NC-p38i.

We have also analyzed how NC-p38i compounds affect the levels of phosphorylation of both p38 α and TAB1 in cells exposed to SIR. In these experiments, we did not observe reduced phospho-p38 levels upon treatment with NC-p38i (see Reviewer Fig. 1 below). However, the levels of TAB1 phosphorylation on S423 were consistently reduced in cells treated with NC-p38i (Reviewer Fig. 1). Since phosphorylation of this residue by p38 α has been linked to the TAB1-induced activation of p38 α during SIR², our results are consistent with the idea that NC-p38i compounds impair the TAB1-induced p38 α autophosphorylation. These observations establish a correlation between decreased p38 α activity, as determined by the diminished phospho-TAB1 levels, and the reduction in cell death upon treatment with NC-p38i compounds.

The observation that p38 α phosphorylation is not reduced when SIR-exposed cells are treated with NC-p38i compounds is surprising, but it is consistent with previous work showing that hearts expressing a TAB1 mutant unable to dock onto p38 α showed reduced TAB1 phosphorylation but only a mild attenuation in p38 α phosphorylation upon *in vivo* myocardial ischemia, which was proposed to be due to compensatory mechanisms for p38 α phosphorylation, for example mediated by MAP2Ks². Our results predict that while NC-p38i compounds do not interfere with the binding of TAB1 to p38 α , they will impair p38 α activation therefore affecting TAB1 phosphorylation. Thus, the work using genetically modified mice² and our own results using NC-p38i compounds both suggest the potential importance of TAB1 phosphorylation for SIR-induced cell injury, but further work is required to characterize the contribution of TAB1 phosphorylation to this process. Since this information does not add much to the characterization of NC-p38i compounds, we would prefer to leave it as a reviewer figure rather than including it in the manuscript.

Reviewer Fig. 1. H9c2 cells were untreated (Control) or treated for 2 h with simulated ischemia followed by 15 min of reperfusion (SIR), in the presence of DMSO (D), the ATP-competitive inhibitor SB203580 (SB, 10 μ M) or the indicated compounds at 30 μ M. The compounds were added 24 h before and maintained during the treatment. Total cell lysates were analyzed by immunoblotting with the indicated antibodies. The histogram shows the quantification of phospho-TAB1 levels normalized to the DMSO-treated cells. Data are shown as the mean \pm SD (n=3 experiments).

Reviewer #3:

The authors describe the identification of a new class of ligands they suggest is able to block p38a autophosphorylation but poorly affects the activity of the canonical pathway. The discovery of p38 pathway-specific inhibitors has the potential to be of high impact, substantial evidence in the medical literature shows that blanket pharmacological inhibition of p38 activity, by ATP-competitive inhibitors causes adverse side effects most likely due to on target effects; the development of inhibitors that are circumstance specific could help overcome the side effects observed with blanket inhibition.

The manuscript is well written, and the experiments are carried out to a high standard. A large number of experiments is presented both in cells and in vitro, but overall, in my opinion, they do not constitute sufficient evidence to support the conclusion that the ligands are selective in inhibiting p38 signalling through autophosphorylation. The difference they observe between transphosphorylation and autophosphorylation inhibition could be simply down to the fact that the enzyme substrate ratio will be significantly different between the two types of activation; in the autophosphorylation reaction p38 is both substrate and enzyme whereas in the transphosphorylation reaction the substrate is in large excess compared to the enzyme. For instance, it is not clear why they use molarity unit- measures when reporting the concentration of the reagents in the autophosphorylation reaction and instead weight/volume when reporting the transphosphorylation reaction. A major issue might be that in the transphosphorylation reaction they are operating at a kinase concentration that is too low for the inhibitors to interact, therefore making the comparison not reliable. At the moment given the different unit measures used it is not possible to compare.

We apologize for the inconsistencies. We have now homogenized the units used for protein concentration throughout the manuscript for better clarity. In addition, we have repeated the autophosphorylation and transphosphorylation reactions using the same p38 α concentrations in both cases to rule out that this could somewhat affect the inhibitory properties of the NC-p38i compounds, as suggested by the reviewer. The new experiments further corroborate that NC-p38i significantly reduce p38 α autophosphorylation but barely

affect the phosphorylation of the ATF2 and MK2 substrates by MAP2K-activated p38 α (new Supplementary Fig. 2a and 2b).

The structural data do not provide sufficient insight into the mechanism through which the ligands might specifically inhibit autophosphorylation. An explanation, at a molecular level, of the selectivity of the ligands might help support their hypothesis. I would suggest the authors check whether ligand binding affects the formation the h-bond between Thr185 and Asp150, which is crucial for the autophosphorylation reaction; mutations that affect the h-bond formation selectively abolish autophosphorylation (PDB:5O90).

Thank you for this comment. Indeed, as suggested by the reviewer, the hydrogen bond between Thr185 and Asp150 cannot be found in any of the p38 α -NC-p38i structures described in our manuscript (new Fig. 7), confirming that the activation loop is positioned away from the active site. The absence of this hydrogen bond, which was previously reported to play a key role in TAB1-mediated p38 α autophosphorylation ¹, as well as the ability of the compounds to interfere with ATP binding to the non-phosphorylated p38 α , both probably contribute to hinder the autophosphorylation process.

Although the hydrogen bond is absent in the three structures described in our paper, we noticed two different scenarios as depicted in the new Fig. 7. When the NC-p38i molecules are located in the active site (7Z9T.A, pink, and 7PVU.A, purple), the region comprising the residues 182-185 does not fold as a helix. In consequence, the Thr185 and Asp150 side chains are positioned far away from each other and cannot form a hydrogen bond. However, when NC-p38i compounds are located in the hinge pocket (7Z6I, green) or in the allosteric pocket (7Z9T.B, cyan), we observe the presence of the helix but not of the expected hydrogen bond. In this case, even though the Thr185 and Asp150 side-chains are at 4.4 Å, far from the ideal hydrogen bond distance and unable to form a hydrogen bond due to their geometrical disposition. For comparison, the figure also includes the structure of p38 α bound to TAB1 described in the literature ³, displaying the above-mentioned helical structure and the hydrogen bond (2.5 Å).

These observations confirm that binding of NC-p38i molecules to either the active or the allosteric sites interferes with the formation of the Thr185-Asp150 hydrogen bond, supporting the specificity of these compounds to selectively abolish the autophosphorylation of p38 α .

It is also not clear what is the binding stoichiometry of the interaction between the ligands and the kinase. The authors write that the structures they solved reveal that NC-p38i compounds predominantly occupy three different areas around the active cavity. What do they mean by predominantly?

We have reported the three sites observed in crystals, but cannot exclude other less stable or transient binding modes that are not durable enough to crystallize. This is the reason why we used the term "predominantly" but we agree that this was not a fortunate choice. To avoid confusion for the readers, we have rephrased this sentence in the revised version of the manuscript as follows: "These structures revealed that NC-p38i compounds occupy three different areas around the active cavity,"

Can they please report whether they observe any extra unassigned electron density in their maps that could be attributed to the ligands?

We did not observe any additional electron density in our maps that could be attributed to ligands. Furthermore, we used the "Validate-> Unmodeled blobs" Coot option, which did not report any unmodeled blobs in the 2mFo-DFc map for blobs even above the RMSD of 1.0. Finally, in regions where water appeared to form chains, we generated Polder maps to see if they could be ligands in place. This procedure revealed no additional ligands and confirmed that the density belonged to water molecules. See Reviewer Fig. 2.

Reviewer Fig. 2. Example of a density cluster initially tested as ligand #37 using polder maps but finally resolved as water molecules. PDB 7Z6I.

The structures of the NC-p38i compounds in complex with p38 were obtained through co-crystallization experiments. Based on the 1:1.5 ratio they report, the protein concentration is between 0.2-0.4mM whereas the NC-p38i compounds are at 0.3-0.6mM, whereas for SB203580 (a much stronger binder) they use a larger excess (1mM), what is the rationale for their choice. Moreover the authors should comment on whether the crystal packing has any effect on the NC-p38i compounds binding mode they observe.

Exploratory experiments before starting the crystal trials, revealed that both NC-p38i compounds have limited solubility at concentrations higher than the ones we used. For SB203580 we used the crystallization conditions previously reported ³.

Before solving the structures of the complexes, we could not be certain about the final location of our compounds. We thought that using either SB203580 or ATP- γ -S would enhance the possibility of getting diffracting crystals, and also of observing our compounds bound to the protein. The reason for using two different competing binders (SB203580 and ATP- γ -S) was that ATP- γ -S has a lower affinity than our NC-p38i molecules for the catalytic site, and we thought that if they shared the same binding site, ATP- γ -S could be displaced by our molecules. In the case of SB203580, we reasoned that this strong binder would occupy the ATP binding site, allowing us to explore possible secondary binding sites of NC-p38i compounds. The combination of these two approaches allowed us to describe the possible binding modes more completely.

In fact, we observe that NC-p38i compounds cannot displace the high affinity inhibitor SB203580 from the ATP binding site, and therefore bind at the hinge. On the other hand, when we use the non-hydrolysable ATP- γ -S analog, our compounds bound to the catalytic site and its proximity.

With respect to crystal packing effects, in the 7PVU and 7Z9T complexes, the ligands are located in or around the catalytic site, away from any symmetry partners, so their positions are unlikely influenced by crystal packing. In the 7Z6I structure, in which the catalytic site is

occupied by the SB203580 molecule, compound #37 is located at the interface of two proteins. See Reviewer Fig. 3.

Reviewer Fig. 3. C-alpha representation of the packing observed in the 7Z6I structure. Compound #37 has been found close to a symmetry neighbor and opposite to the ATP binding site where the SB203580 is located. Thr180 in the activation loop is labeled to show how this loop extends far from the active center.

In the protein expression section of the manuscript, it is described how the cells expressing His-SUMO-p38a protein are grown in deuterated water and minimal media, what is the reason for this?

We prepared labeled proteins to test by NMR if they were properly folded before starting the crystallization trials. We have added this sentence to the revised Methods section.

Finally the authors should report the CC values for the polder maps in Fig. S6.

We have added the CC values for Polder maps to the new Supplementary Fig. 7 (old Figure S6). In addition, we have explained how we validated the Polder maps in the Methods section:

"Validation of polder maps was performed following the approach described in ⁴. All ligands reported using polder maps in this paper have CC(1,3) values larger than CC(1,2) and CC(2,3), which is the acceptance criterion."

Regarding the Editor's request to make sure that omit maps are provided for our ligand-bound structures, we did not use omit maps but positive difference maps from Molecular Replacement. The NC-p38i compounds were found using polder maps as explained above. The SB2023580 ligand was located immediately after Molecular Replacement of an only-protein model. We have deposited all maps at the Worldwide Protein Data Bank (Supplementary Table 4).

References

1. Thapa, D. et al. TAB1-Induced Autoactivation of p38alpha Mitogen-Activated Protein Kinase Is Crucially Dependent on Threonine 185. *Mol Cell Biol* **38**(2018).
2. De Nicola, G.F. et al. The TAB1-p38alpha complex aggravates myocardial injury and can be targeted by small molecules. *JCI Insight* **3**, e121144 (2018).
3. DeNicola, G.F. et al. Mechanism and consequence of the autoactivation of p38alpha mitogen-activated protein kinase promoted by TAB1. *Nat Struct Mol Biol* **20**, 1182-90 (2013).
4. Liebschner, D. et al. Polder maps: improving OMIT maps by excluding bulk solvent. *Acta Crystallogr D Struct Biol* **73**, 148-157 (2017).

REVIEWERS' COMMENTS

Reviewer #2 (Remarks to the Author):

This revision provided additional information to further support the mechanism of action for the novel p38i on TAB1 mediated autophosphorylation. The revised manuscript also highlighted some of the remaining questions and more proper interpretations. Overall, the manuscript is interesting and high quality. The content can add interesting and useful insights to p38 regulation and inhibitory mechanism of the new compounds.

Reviewer #3 (Remarks to the Author):

The authors have addressed my concerns I am happy for the manuscript to be published.

Manuscript NCOMMS-22-42583-T. Response to Reviewers.**Reviewer #2 (Remarks to the Author):**

This revision provided additional information to further support the mechanism of action for the novel p38i on TAB1 mediated autophosphorylation. The revised manuscript also highlighted some of the remaining questions and more proper interpretations. Overall, the manuscript is interesting and high quality. The content can add interesting and useful insights to p38 regulation and inhibitory mechanism of the new compounds.

Thank you.

Reviewer #3 (Remarks to the Author):

The authors have addressed my concerns I am happy for the manuscript to be published.

Thank you.